# Incremental model breakdown to assess the multi-hypotheses problem

Florian U. Jehn[1], Lutz Breuer[1,2], Tobias Houska[1], Konrad Bestian[1], Philipp Kraft[1]

[1]Institute for Landscape Ecology and Resources Management (ILR), Research Centre for BioSystems, Land Use and Nutrition (iFZ), Justus Liebig University Giessen, Heinrich-Buff-Ring 26, 35390 Giessen, Germany
[2]Centre for International Development and Environmental Research (ZEU), Justus Liebig University Giessen, Senckenbergstrasse 3, 35392 Giessen, Germany

*Correspondence to*: Florian U. Jehn (florian.u.jehn@umwelt.uni-giessen.de)

**Abstract.** The ambiguous representation of hydrological processes have led to the formulation of the multiple hypotheses approach in hydrological modelling, which requires new ways of model construction. However, most recent studies focus only on the comparison of predefined model structures or building a model step-by-step. This study tackles the problem the other way around: We start with one complex model structure, which includes all processes deemed to be important for the catchment. Next, we create 13 additional simplified models, where some of the processes from the starting structure are disabled. The performance of those models is evaluated using three objective functions (logarithmic Nash-Sutcliffe, percentage bias and the ratio between root mean square error to the standard deviation of the measured data). Through this incremental breakdown, we identify the most important processes and detect the restraining ones. This procedure allows constructing a more streamlined, subsequent 15th model with improved model performance, less uncertainty and higher model efficiency. We benchmark the original Model 1 and the final Model 15 with HBV-Light. The final model is not able to outperform HBV-Light, but we find that the incremental model breakdown leads to a structure with good model performance, fewer but more relevant processes and less model parameters.

## 1 Introduction

In the world of hydrological modelling, scientists construct models and apply them for a specific research question. Sometimes, these models are modified or extended afterwards, but the core components stay the same. This approach has existed from the earliest days of simple equations until the models of connected, conceptual elements used today (Todini, 2007).

During the development of hydrological models, the issues of parameter and input data uncertainty were often in the center of the scientific debate and numerous methods for assessing this uncertainty have been proposed. Structural uncertainty has been investigated in the past decade (Breuer et al., 2009; Son and Sivapalan, 2007) and gained more momentum in the last few years (e.g Clark et al., 2015; Fenicia et al., 2011; Hublart et al., 2015). It was noted that problems often arose from the focus on trying to build one model that was meant to work equally well for all catchments (Fenicia et al., 2011).

In order to better scrutinize problems associated with the model structure, the theory of the multiple hypotheses was introduced, first by Beven (2001, 2002), and more recently picked up by Clark et al. (2011). This theory enables a more structured approach to model building, as it identifies a given model not as a single hypothesis, but as an assemblage of coupled hypotheses. Hence, Clark et al. (2011) proposed that a model should be constructed in a way that allows the testing of every single hypothesis of every process separately. In addition, the interactions of single elements within such a model should also be considered to better understand why a certain model works or fails (Clark et al., 2016).

When the idea of multiple hypotheses emerged, there was no easy way to construct models with interchangeable components (Buytaert et al., 2008) except for some comparison inside the TOPMODEL model family (Beven and Kirkby, 1979). However, Bergström (1991) already noted that a critical view on the necessity of process descriptions is important in the development of models like HBV. We now have model frameworks at hand that facilitate such a design, e.g. SUPERFLEX (Fenicia et al., 2011), Structure for Unifying Multiple Modelling Alternatives (SUMMA) (Clark et al., 2015b, 2015a), or the Catchment Modelling Framework (CMF) (Kraft et al., 2011). SUPERFLEX targets the construction of lumped conceptual models (van Esse et al., 2013; Gharari et al., 2014). SUMMA and CMF support the generation of multi scale approaches from plot over hillslope to basins and from lumped to fully distributed models. SUMMA focusses on the comparison of process-based models with predefined parameters sets and is up to now mainly tested for surface-atmosphere interactions (Clark et al., 2015b, 2015a). CMF is a programming library to build hydrological models from building blocks with both, process-based and conceptual models. It can be used for subsurface and surface water fluxes, surface-atmosphere exchange and solute transport. So far, it has been applied in studies to better understand hydrological processes (Holländer et al., 2009; Maier et al., 2017; Orlowski et al., 2016; Windhorst et al., 2014), to simulate solute transport (Djabelkhir et al., 2017; Kraft et al., 2010) and to capture hydrological lateral and vertical transport processes in coupled complex ecosystem models (Haas et al., 2013; Houska et al., 2014, 2017; Kellner et al., 2017).

All toolboxes enable a stepwise modification of the model structure. Additionally, they allow an easier comparison of different models, as they are all constructed from the same parts and a more straightforwardly handled through interfaces (Buytaert et al., 2008). Recently, some studies tried to tackle the multi-hypotheses problem within a model framework (e.g. van Esse et al., 2013; Fenicia et al., 2008; Gharari et al., 2014; Hublart et al., 2015; Kavetski and Fenicia, 2011). Most of these studies built their models incremental from bottom up to find out, if small modifications allow a better simulation (Bai et al., 2009; Westerberg and Birkel, 2015). Others compared predefined model structures (van Esse et al., 2013; Kavetski and Fenicia, 2011). In all cases, researchers stopped improving the models once a sufficient performance was reached. Clark et al (2015ab) propose another approach to test multiple hypotheses. Here, the number and type of subprocesses stay static, yet the mathematical formulation of the process descriptions are scrutinized by exchange. However, the different model scrutinizing strategies are independent of the selected framework. SUMMA, SuperFLEX, FUSE and CMF should be suitable for incremental model build-up, process exchange as well as our new proposal of the incremental model break down strategy.

Despite having the potential to create a wide range of models with such toolboxes, only a minor quantity in the vast space of possible model structures is currently explored. However, this thorough exploration is needed to find appropriate model

structures for any catchment, as it seems that current hydrological knowledge does not allow to construct a model that works equally well for all environmental conditions, especially when using lumped models (Beven, 2000, 2007, 2016; Buytaert et al., 2008; Fenicia et al., 2014).

To better use the existing understanding of a given catchment and to test more complex models, this study turns the incremental approach of adding more process-understanding to a model upside down. First, we develop a conceptual model from current hydrological understanding that contains all subprocesses that might be important for the functioning of a catchment. Then, parts of this model are disabled through incremental model breakdown, and the reduced model structures are tested for their simulation performance. A subprocess is marked as necessary, when models lacking it are rejected. On this base, a subsequent model is constructed which uses only meaningful subprocesses. Incremental model breakdown is therefore a rejectionist approach, built on the learning from failure and not an optimization process. Beven (2006) assumed that a rejectionist approach is generally better suited to gain insight about process hypotheses. To allow comparability of the incremental breakdown method with common modelling approaches, the subsequent model is finally benchmarked with HBV-Light.

The objective of this study is to demonstrate that incremental model breakdown allows a detailed examination of model structures, an easier identification of the most important hydrological processes, and thus the construction of an improved model. While still not being able to sample the entire space of possible model structures, this approach might find some model structures which are likely missed with other methods. Ultimately, this approach also enables a better hydrological understanding of the catchment, as different structures, flaws and errors of a first modelling approach become obvious, even if a theoretical optimal model structure is still unknown..

## 2 Material and Methods

### 2.1 Study area

The study area is an upper section (AEO 2,977 km$^2$, gauging station Grebenau) of the Fulda catchment (Figure 1), a catchment with Mid-European temperate climatic conditions. Relevant processes and catchment characteristics to be considered included the contribution of snowfall to precipitation, a mix of land uses with open and closed vegetation cover, and urban regions that impact hydrology through non-gradient driven fluxes (e.g. water abstraction for drinking water supply, sewage treatment works, reservoirs or sealed areas).

Precipitation input is influenced by the surrounding low mountain ridges of the Vogelsberg, the Wasserkuppe, the Knüll-Mountains and the Melsunger Uplands, leading to a significant contribution of snowfall in winter. The elevation ranges from about 150 m a.s.l. at Grebenau to 950 m a.s.l. at the Wasserkuppe. Wittmann (2002) used tritium as a tracer and found that the Fulda catchment has two distinct groundwater reservoirs: A large one reacting slowly and a smaller one with faster reaction. Land use is dominated by agriculture (37 %) and forests (41 %).

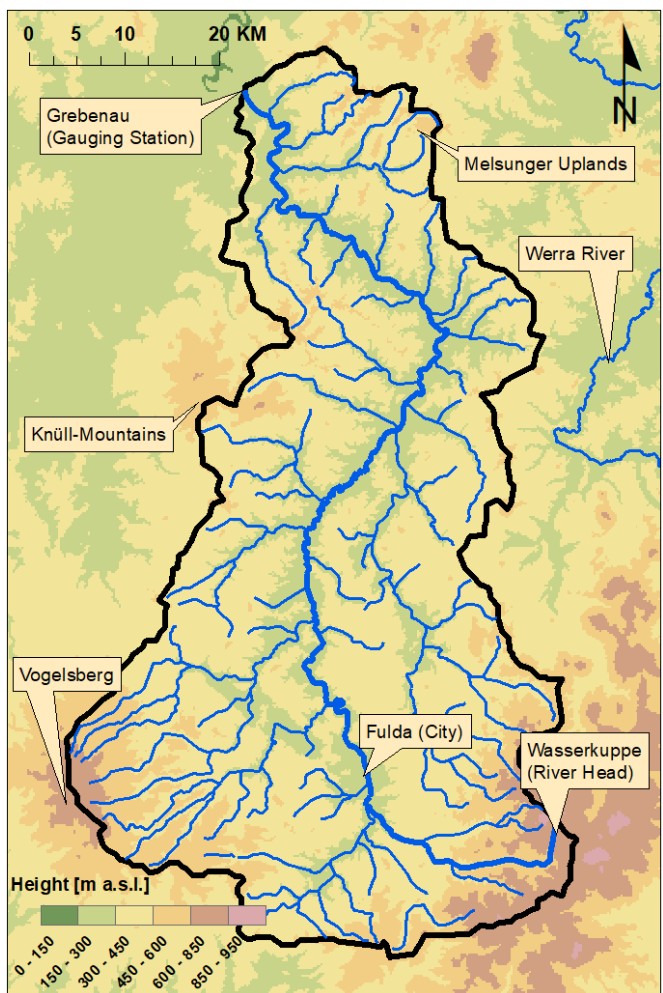

**Figure 1: Relief map of the partial catchment of the Fulda River (black border) with side streams, ridges and parts of the Werra River.**

## 2.2 Model input and validation data

Discharge data for the gauging station Grebenau, temperature and precipitation data were obtained from the Hessisches Landesamt für Naturschutz, Umwelt und Geologie (HLNUG). The point measurements for precipitation and temperature of the 51 measurement stations were extrapolated over the whole catchment, using kriging with altitude as an external drift (Hudson and Wackernagel, 1994). Finally, the extrapolated values were averaged over the whole catchment to get a single, lumped value per day. The time period from 1979 to 1988 was choosen because External Drift Kriging requires a high density of precipitation stations and this mentioned period covered most measuring stations with continuous data.

The Fulda catchment has a humid, temperate climate, with an annual precipitation of 838 mm. The annual runoff coefficient ranges between 0.3 to 0.6 (average 0.39), which is in the range for comparable catchments (e.g. Rawlins et al., 2006). The discharge and groundwater of the catchment are influenced by drinking water abstraction for 80,000 inhabitants (Rhönenergie Fulda GmbH, 2017).

The model time step and temporal resolution of the data are both daily, which is sufficient given the size of the catchment and the respective rainfall-runoff response (Sikorska and Seibert, 2018). However, the method presented in this study and the utilized modelling software is not limited to the time step of the forcing or calibration data, but can also be applied on higher resolution data with likely different results. Both the validation and the calibration period behave differently in regard of their patterns of precipitation and discharge (Figure 2). The calibration period is wetter and contains six of the seven large rainfall

events (>30 mm d$^{-1}$) are located here. In addition, in both periods there is one year which represents more extreme weather conditions: 1985 for the calibration period with very little discharge in comparison with the precipitation and 1988 with very much discharge in comparison to the precipitation. Still the precipitation stays in the long term range for this catchment for all years (Fink and Koch, 2010). The exact calibration and validation scheme is explained in 2.5.

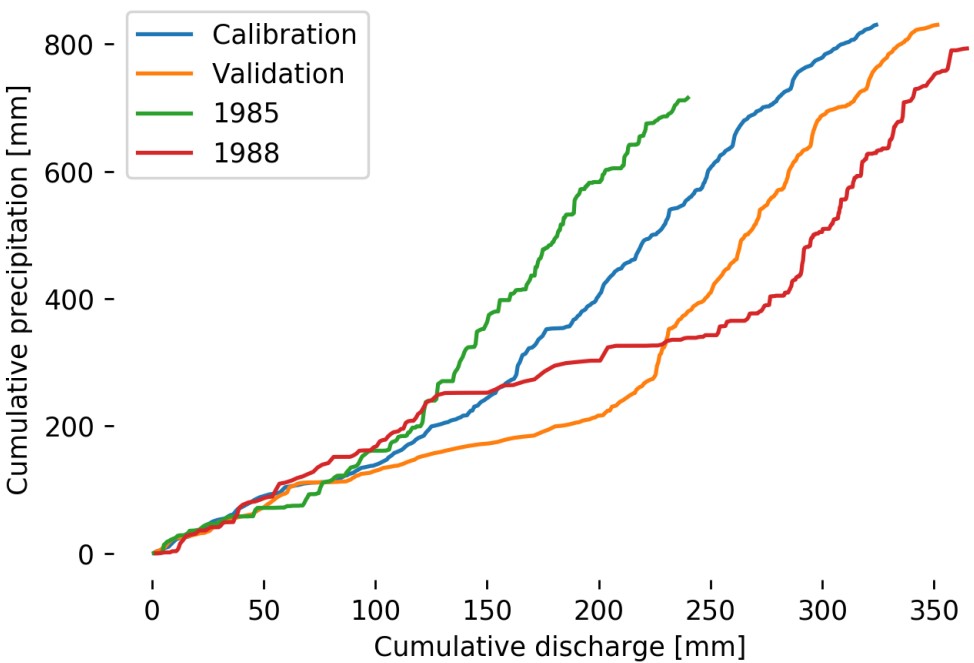

**Figure 2: Cumulative discharge plotted against cumulative precipitation for the calibration (1980 – 1985) and validation (1986 – 1988) period and two years which deviate most from the other years. For the calibration and validation period, the cumulative discharge and precipitation are the average of the corresponding years.**

## 2.3 Model development using Catchment Modelling Framework (CMF)

For the construction of all models and all numerical calculations (except HBV-Light), we used CMF. CMF is a modular framework for hydrological modelling developed by Kraft et al. (2011) (see also CMF, 2017). For solving the differential equations of models constructed with CMF, several numerical solvers are embedded in the toolbox. To avoid numerical problems (Clark and Kavetski, 2010; Kavetski et al., 2011; Kavetski and Clark, 2011) we selected the CVode Integrator (Hindmarsh et al., 2005) for all models. The CMF version used for this study was 0.1380.

In a first model set up (Model 1, Figure 3) all processes are reliant on different flow connections. The incoming precipitation is saved in a snow storage in case the air temperature is below freezing point and rereleased to the surface storage after snowmelt. All other precipitation is split between the canopy or reaches the surface directly, depending on canopy closure. From the surface, the water is either directly routed to the river or enters three serial soil/groundwater layers, which in turn route water to the river as well. In addition, a fixed amount of water is abstracted from the lower groundwater to simulate drinking water extraction, which in turn is routed to the river. The river then routes all water to the outlet. Thus, it contains the implementations of processes for evapotranspiration, a canopy, snow, surfaces, a river, upper- and lower groundwater body (Figure 3).

Following the findings of Singh (2002) all connections in the model with a flow curve (Figure 3) are described as kinematic waves (Equation 1) (Singh, 2002), except for the infiltration and the drinking water abstraction.

$$Q = Q_0 \left(\frac{V - V_{residual}}{V_0}\right)^\beta \tag{1}$$

where Q is the ratio of transferred water, $V_{residual}$ [m³] is the volume of water remaining in the storage, $V_0$ [m³] is the reference volume to scale the exponent, V is the current volume of water in the storage [m³], and $\beta$ is a parameter to shape the response curve [-]. The parameter $Q_0$ is the flux in [m³ d⁻¹], when $\frac{V - V_{residual}}{V_0} = 1$.

Water that reaches the surface, i.e. throughfall or snowmelt, is routed into the upper soil as infiltration, with the following limits applied:

- Infiltration excess expressed by a maximum surface permeability $K_{sat}$
- Saturation excess expressed by a limiting factor calculated from the water content of the first subsurface water storage using a sigmoidal function

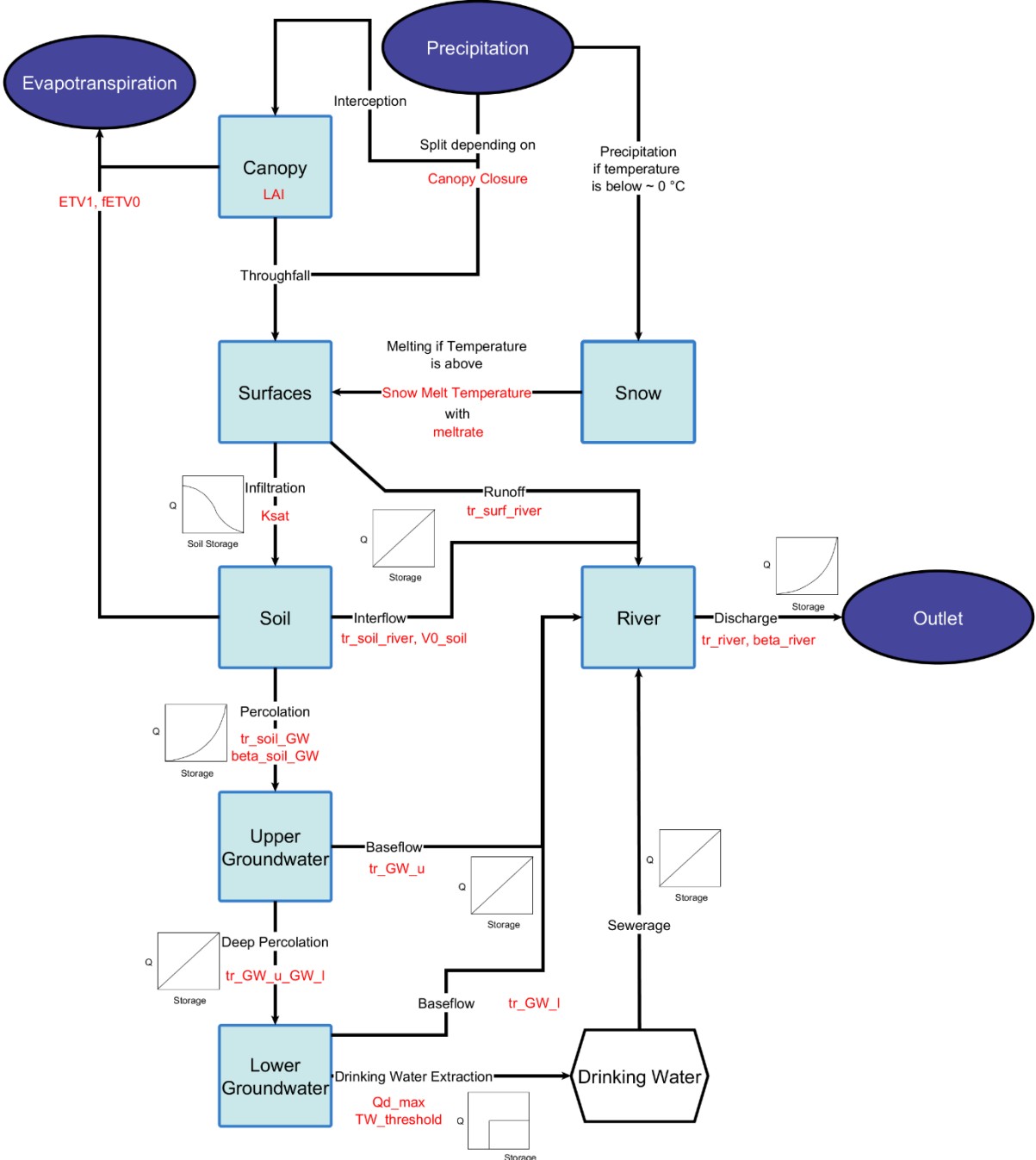

**Figure 3: Structure of the Model 1 with water storages (light blue), boundaries (dark blue), temporary storages (white) calibrated parameters (red), fluxes (black arrows) and flow curves (for all applicable fluxes). Water reaching the outlet is shifted one day into the future.**

Drinking water abstraction is implemented as a fixed amount of water. As the influence of the drinking water abstraction is not known, the amount of water abstracted is calibrated. It is transferred to the drinking water storage as long as the amount of water in the groundwater storage is above a threshold. As this threshold is not exactly known, we included it as a parameter for calibration. From the drinking water storage, all water abstracted for a given day is routed to the river.

Snowmelt uses a simple degree-day method (see API in CMF (2017)). The snowmelt temperature parameter was calibrated. Interception from the canopy is realized as Rutter interception (Rutter and Morton, 1977). Potential evapotranspiration was calculated with the modified Hargreaves equation by Samani (2000).

The devised model was tested by using fluxogram graphs. Fluxograms allow creating animated graphs that resemble model structures with all their storages and fluxes and they were used to analyse the implemented model processes (for more information about the fluxogram graph see Jehn (2018)). The size of the storages and fluxes change for each time step according to the amount of water stored/moved. For the fluxogram animations, we used the model with the highest logarithmic Nash-Sutcliffe Efficiency (logNSE).

## 2.4 Incremental model breakdown

To test the influence of different structure elements in Model 1, we used the concept of a one-at-a-time sensitivity analysis, i.e. disabling one process after the other, to track changes (Figure 4). This resulted in 13 additional models with varying disabled processes (Table 1). In a second step, we disabled up to four processes, to scrutinize the interplay of processes, as proposed by Clark et al. (2011). When a process was removed, the connections leading to it were then connected to the next nearby storage. For example: If the surface storage was removed, the Canopy and the Snow Storage were connected to the soil. Or if the river was removed, all connections leading to it were directly connected to the outlet.

For each simplified model, the explanatory model performance during the calibration period was evaluated and the validation period was not used in any model parameter / structure selection process but spared for the a posteriori model comparison. . If the model performance was getting worse, the deleted process was valued essential for the model and vice versa. If the performance did not change, the process was rated unimportant. This allowed us to separate influential processes from unnecessary ones and thereby assess if the chosen complexity was justifiable for the catchment (Figure 5, Figure 6). The main criteria to determine the value of a process was the ability of the model to produce behavioral runs in the calibration period at all. A model was rejected, when it is not able to produce runs of acceptable performance for all parameters. And rejected means in this study, the model is missing a process too important to ignore. If a model lacking a certain subprocess is able to produce behavioural runs that subprocess is irrelevant for this application. The final Model 15 was constructed from Model 1 by removing the irrelevant subprocesses

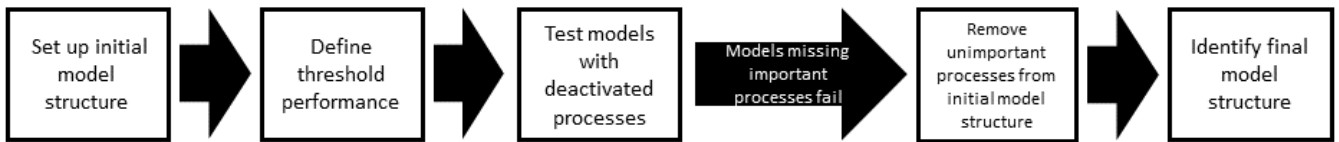

**Figure 4: Flow chart for the method of incremental model breakdown**

**Table 1: Structural elements of the models and amount of parameters. GW = Groundwater, DW = Drinking Water. ET = Evapotranspiration. Light gray indicates active components. Dark grey indicates disabled components.**

| | Rain distribution | | | | | Groundwater (GW) | | | | Number |
| | Canopy | Surfaces | Snow | Soil | River | upper GW | lower GW | DW | ET | Parameters |
|---|---|---|---|---|---|---|---|---|---|---|
| Model 1 (start) | | | | | | | | | | 19 |
| Model 2 (no GW) | | | | | | | | | | 13 |
| Model 3 (no ET) | | | | | | | | | | 18 |
| Model 4 (no river) | | | | | | | | | | 17 |
| Model 5 (no rain distribution) | | | | | | | | | | 13 |
| Model 6 (no surfaces) | | | | | | | | | | 17 |
| Model 7 (no canopy) | | | | | | | | | | 17 |
| Model 8 (no snow) | | | | | | | | | | 17 |
| Model 9 (no DW) | | | | | | | | | | 17 |
| Model 10 (no GW/river) | | | | | | | | | | 10 |
| Model 11 (no canopy/surfaces) | | | | | | | | | | 15 |
| Model 12 (no river/surfaces) | | | | | | | | | | 15 |
| Model 13 (no lower GW) | | | | | | | | | | 17 |
| Model 14 (no lower GW/DW) | | | | | | | | | | 15 |
| Model 15 (final) | | | | | | | | | | 10 |

## 2.5 Calibration and validation

10   A model run was separated into warm-up period of one year (1979), a calibration period of six years (1980-1985) and a validation period of three years (1986-1988). First CMF model runs showed that many simulated discharge peaks occurred one day ahead compared to observed data. This is caused by rainfall occurring in the later time of a day, that leads to a reaction in the hydrograph of the following day as water needs time to reach the gauging station (Ficchì et al., 2016). The model, however, reacts directly to this as its input data is resolved in a 24 h time step. Therefore, we shifted the simulated time series

15   one day into the future as proposed by Bosch et al. (2004). This led to better calibration results. This was not needed for HBV-Light, in which the MAXBAS parameter accounts for shifting peak discharge.

We used the Generalized Likelihood Uncertainty Estimation (GLUE) methodology (Beven and Binley, 1992) to find behavioral parameters sets for the calibration period. It should be noted, that other calibration schemes, objective functions and parameter ranges might have led to different results. However, we are not striving to find the best performing parameter set. Instead, we uses GLUE for the identification of behavioral model runs to evaluate the various model structures. As single-objective calibration lowers the identifiability of model parameters and structural elements (Efstratiadis and Koutsoyiannis, 2010) and often hide shortcomings of models (Ritter and Muñoz-Carpena, 2013), we pursued a multi-objective calibration procedure. Following the concept of Moriasi et al. (2007), a model run was deemed behavioural, if the Nash-Sutcliffe-Efficiency (NSE) was >0.5 (optimal value: 1; range: 1 to - ∞), the percentage bias (PBIAS) was below/above ±25% (optimal value: 0; range: -100 to + ∞) and the ratio between root mean square error to the standard deviation of the measured data (RSR) was <0.7 (optimal value: 0; range: 0 to ∞). As an additional constraint we also included the logarithmic Nash-Sutcliffe-Efficieny (logNSE) (optimal value: 1; range: 1 to - ∞) to allow a better evaluation of low flows. The NSE focuses on peak flows and the PBIAS considers the overall model deviation from observed data. It should be noted though, that this study does not aim on finding the optimal parameter sets for a single model, but to use the knowledge gained from calibration and validation to identify the most important processes in the model structure and use this to improve the model structure and reduce the number of parameters used. The validation period is strictly not used in any selection process to avoid overfitting and only used in the last validation step of the overall method. To finally compare the models, that were able to produce behavioural runs, we used the median to evaluate the typical behaviour of a run of a given model and the maximal value to determine the best possible performance.

The sampling of the parameter space for calibration was done by Latin Hypercube Sampling (McKay et al., 1979) implemented via SPOTPY (Houska et al., 2015). All models were run 300,000 times each, using a High Performance Computing Cluster. See the tutorial section of CMF (2017) for more detailed information on the coupling of CMF with SPOTPY for model calibration. Implemented parameter boundaries for Model 1 are given in Table 2 and remained fixed for all further developed model structures to ensure comparability.

The lower and upper bounds for V0_soil and ETV1 were taken from Blume et al. (2016) for typical field capacities reported for German soils in the range of 20 to 300. Canopy parameters are in line with values provided by Breuer et al. (2003). Groundwater transit times are roughly corresponding with the findings of Wittmann (2002) and Wendland et al. (2011). For all other parameters, we could not find reliable data and thus estimated them subjectively. The parameters use a wide range intentionally to allow the parameters to adapt to the very different model structures.

## 2.6 Benchmarking model

As there have not been many studies regarding the construction of models via modelling frameworks, this study uses HBV-Light as a benchmark to make results more comparable with non-framework studies and to allow a more precise evaluation of

the performance of the proposed incremental model breakdown method. HBV-Light is a widely used model, which has proven its functionality in very diverse catchments [*Seibert and Vis*, 2012]. It is a lumped, parsimonious model. We used the simplest setup of HBV-Light with a single soil storage and no lapse rate. As HBV-Light has no internal way to calculate potential evapotranspiration, we used the same approach by *Samani* [2000] as for all other models.

| Name | Unit | Intended meaning | Min | Max |
|---|---|---|---|---|
| tr_soil_GW | day | Residence time from soil to upper GW | 0.5 | 150 |
| tr_soil_river | day | Residence time from soil to river | 0.5 | 55 |
| tr_surf_river | day | Residence time from surfaces to river | 0 | 30 |
| tr_GW_l | day | Residence time from upper GW to river/outlet | 1 | 1000 |
| tr_GW_u | day | Residence time from upper GW to river/outlet | 1 | 750 |
| tr_GW_u_GW_l | day | Residence time from upper to lower GW | 10 | 750 |
| tr_river | day | Residence time from river to outlet | 0 | 3.5 |
| V0_soil | mm | Field capacity of the soil | 15 | 350 |
| beta_soil_GW | / | Exponent which changes the shape of the flow curve | 0.5 | 3.2 |
| beta_river | / | Exponent which changes the shape of the flow curve | 0.3 | 4 |
| ETV1 | mm | Volume under which the evapotranspiration is lowered | 0 | 100 |
| fETV0 | % | Factor by what the evapotranspiration is lowered | 0 | 0.25 |
| meltrate | mm °C$^{-1}$ day$^{-1}$ | Meltrate of the snow | 0.15 | 10 |
| snow_melt_temp | °C | Temperature of snow melt | -1 | 4.2 |
| Qd_max | mm day$^{-1}$ | Maximal drinking water extraction | 0 | 3 |
| TW_threshold | mm | Amount of water that cannot be extracted | 0 | 100 |
| LAI | / | Leaf area index | 1 | 12 |
| CanopyClosure | % | Canopy closure | 0 | 0.5 |
| Ksat | m day$^{-1}$ | Saturated conductivity of the soil | 0 | 1 |

**Table 2: Lower and upper parameters bounds of all models and their indented meaning. GW = Groundwater**

## 3 Results

### 3.1 Behavioural runs of Model 1 to 14

Model 1 was able to achieve nine behavioural runs. The model has a better performance in the validation period (Figure 5, Figure 6). This is true for all other models as well. The simulated discharge is rather erratic (Figure 7), i.e. it reacts directly on small changes in precipitation. Those quick reactions are timed correctly. However, they overestimate the discharge from small precipitation events, while underestimating large ones. These differences are larger in summer than in winter. This behaviour

leads to underestimated high flows and many overestimated small peaks, while the overall simulated amounts are unbiased. Investigation of storages and fluxes (fluxogram-graph: https://youtu.be/cP0PfDpfW88) show that most of the water is stored in the upper groundwater storage, while the lower groundwater storage is removed directly by the drinking water production, as soon as it is above the threshold. Only very small amounts of water are stored in the surface storage, the canopy storage and the soil storage. From the soil storage and the canopy storage large amounts of water evaporate, often exceeding the flow to the outlet. The river storage is mostly recharged from the groundwater and the drinking water storage. The soil storage contributes significantly to the river storage only at large precipitation events or during snowmelt. Overall, Model 1 slightly overestimates low flows and the evapotranspiration, while largely underestimating the peaks.

Most of the deleted model processes from the most complex Model 1 led to more behavioural runs (Figure 5, Figure 6). Model 1, 4 (no river storage), 6 (no surface storage), 9 (no drinking water simulation), 12 (no river and surface storages), 13 (no lower groundwater storage) and 14 (no groundwater storages and drinking water simulation) have between two to nine behavioural runs. Model 7 (no canopy) and 11 (no canopy and surface storage) are able to produce 80 and 90 behavioural runs respectively (Figure 5, Figure 6). The remaining models 2 (no groundwater storages), 3 (no evapotranspiration), 5 (no rain distribution), 8 (no snow) and 10 (no groundwater and river storages) were not able to produce behavioural runs.

Most, but not all simplified models tend to show better performances for their median values of the logNSE, NSE and the RSR (at least in the validation period) than Model 1, while Model 1 has a PBIAS better than the other models. Especially Model 13 (no lower groundwater storage) median values for the objective functions outperform Model 1.

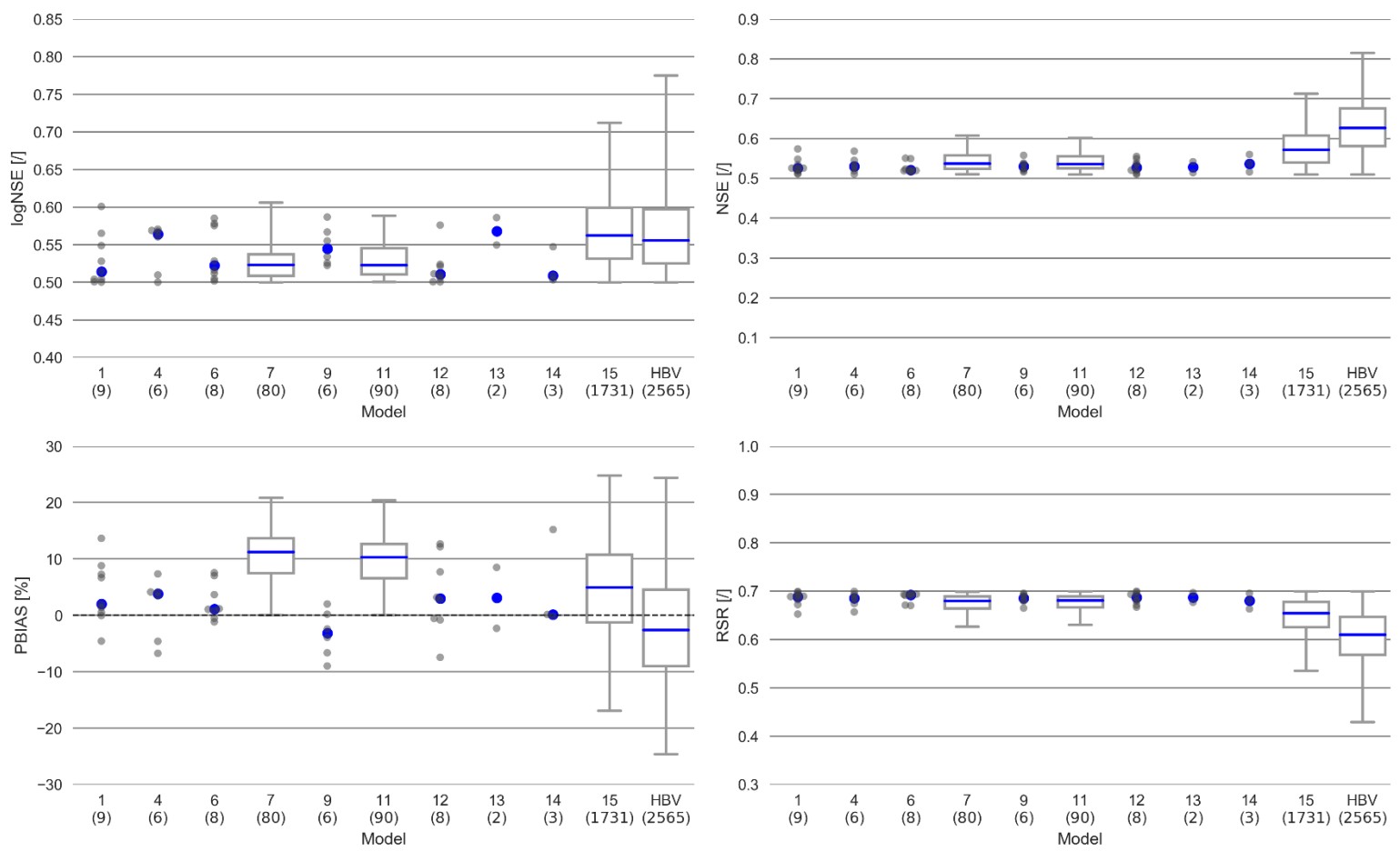

**Figure 5: Distribution of the values of the behavioural runs in the calibration period. Plotted as Boxplot if the number of behavioural runs is above 10. Number of behavioural runs in brackets under the name of the model. Blue dot/line marks the median.**

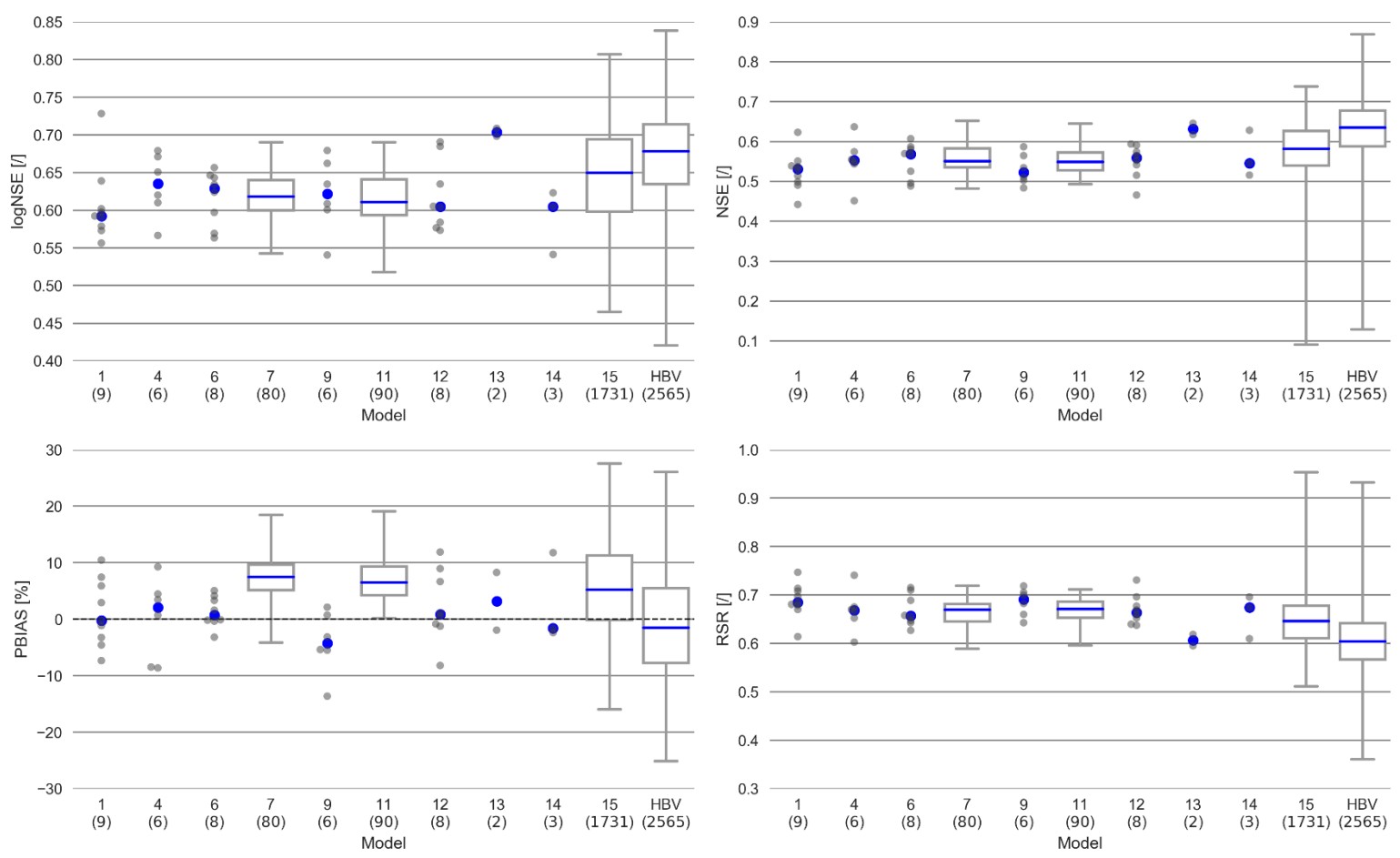

**Figure 6: Distribution of the values of the behavioural runs in the validation period. Plotted as Boxplot if the number of behavioural runs is above 10. Number of behavioural runs in brackets under the name of the model. Blue dot/line marks the median.**

.

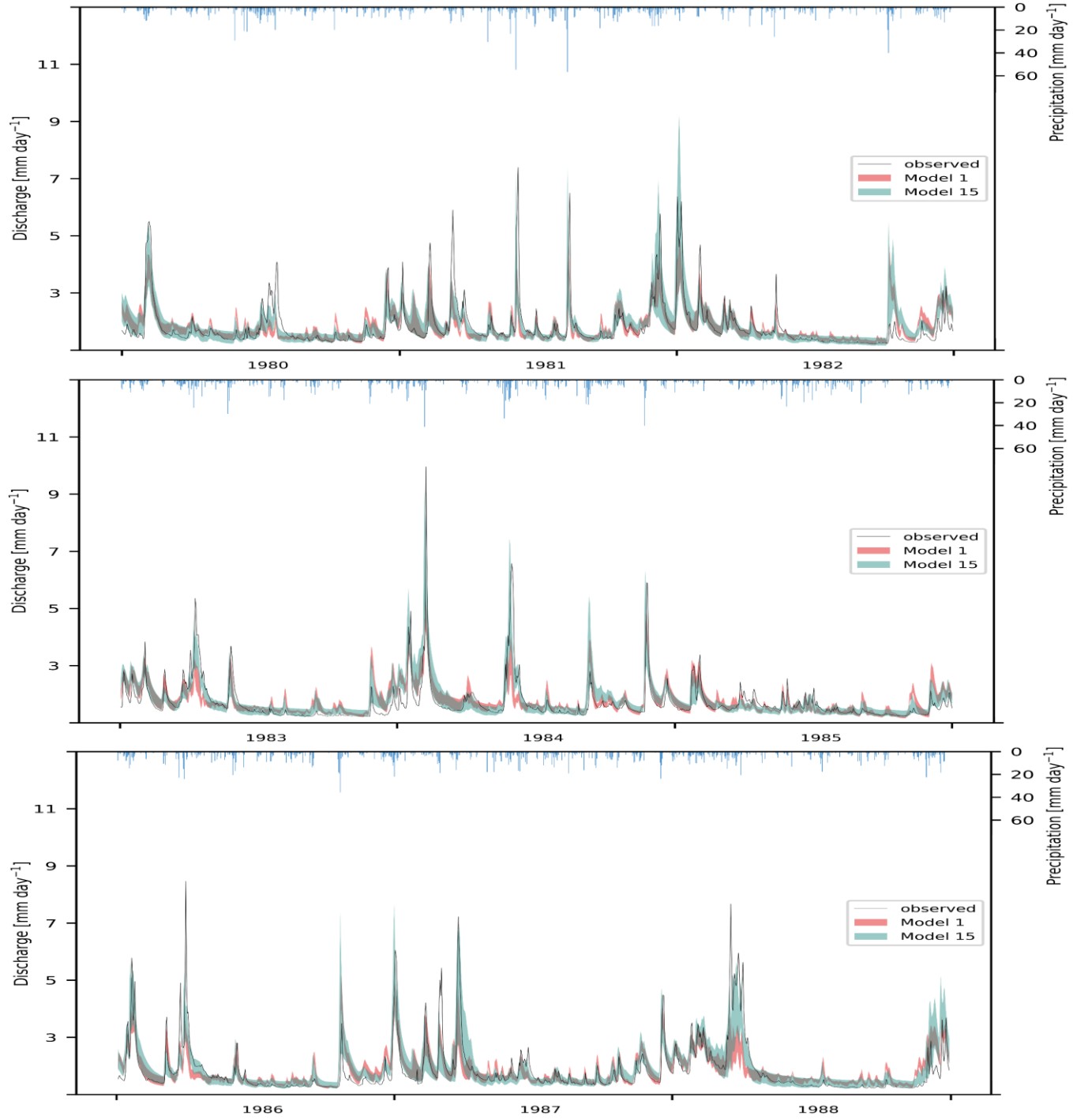

**Figure 7: Hydrograph of the Fulda River for 1980-1988. Coloured areas depict the range of the behavioural model runs (5th to 95th percentile). Calibration period is from 1980 till 1985 (first two subplots). Validation period is from 1986 to 1988 (third subplot). Observed discharge is depicted as black line. Precipitation is drawn with an inverted y-axis.**

## 3.2 Construction and behavioural runs of Model 15

We can report that representation of model processes for the upper groundwater body, evapotranspiration and snow have a positive impact on model performance in the Fulda catchment, given the increased median values of the objective functions (Figure 5, Figure 6), as when those processes are excluded the models struggle to produce behavioural runs. The exclusion of the canopy and drinking water have a more or less neutral impact on the median performance of the behavioural runs (Figure 5, Figure 6). Whereas the chosen implementations of the river, the surfaces and the lower groundwater affect the model quality negatively (Figure 5, Figure 6). The structure of Model 15 was created after all the other models had been evaluated. For this, we used the process knowledge gained from the reduced models (see discussion) and constructed Model 15 with only those processes, which had proven to have positive impact on the quality of the results. Therefore, Model 15 consists only of those processes most important for the given Fulda catchment (Figure 9). In comparison with the model structure of Model 1, the processes surface water storage, lower groundwater storage, drinking water extraction, river storage and the simulation of the canopy were disabled.

Profiting from the insights of the models with disabled processes, Model 15 performs better than Model 1. The RSR, NSE and the logNSE depict better values, both in the validation and calibration period, while the PBIAS is slightly worse for both cases (Figure 5, Figure 6). Especially the maximal values are for the logNSE, NSE and RSR are much better than Model 1. As for all other models, the performance increases from the calibration to the validation period for Model 15. The simulated hydrograph is a lot less erratic than the one from Model 1 (Figure 7). In addition, the peaks fit better than in Model 1. However, summer peaks are less likely to be predicted than those during the rest of the year (Figure 7). The overestimation of low flow in Model 1 is apparent on fewer days. In Addition, to this increase of the performance in comparison with Model 1, Model 15 uses nine parameters less (Table 1). The remaining ten parameters in Model 15 behave differently from the same ones in Model 1 (Figure 8). Some parameters like tr_soil_GW and fEVT0 have almost the same density distribution. Still, there are several parameters like tr_soil_river and ETV1 whose density is much more focused around a specific value for Model 1 than for Model 15.

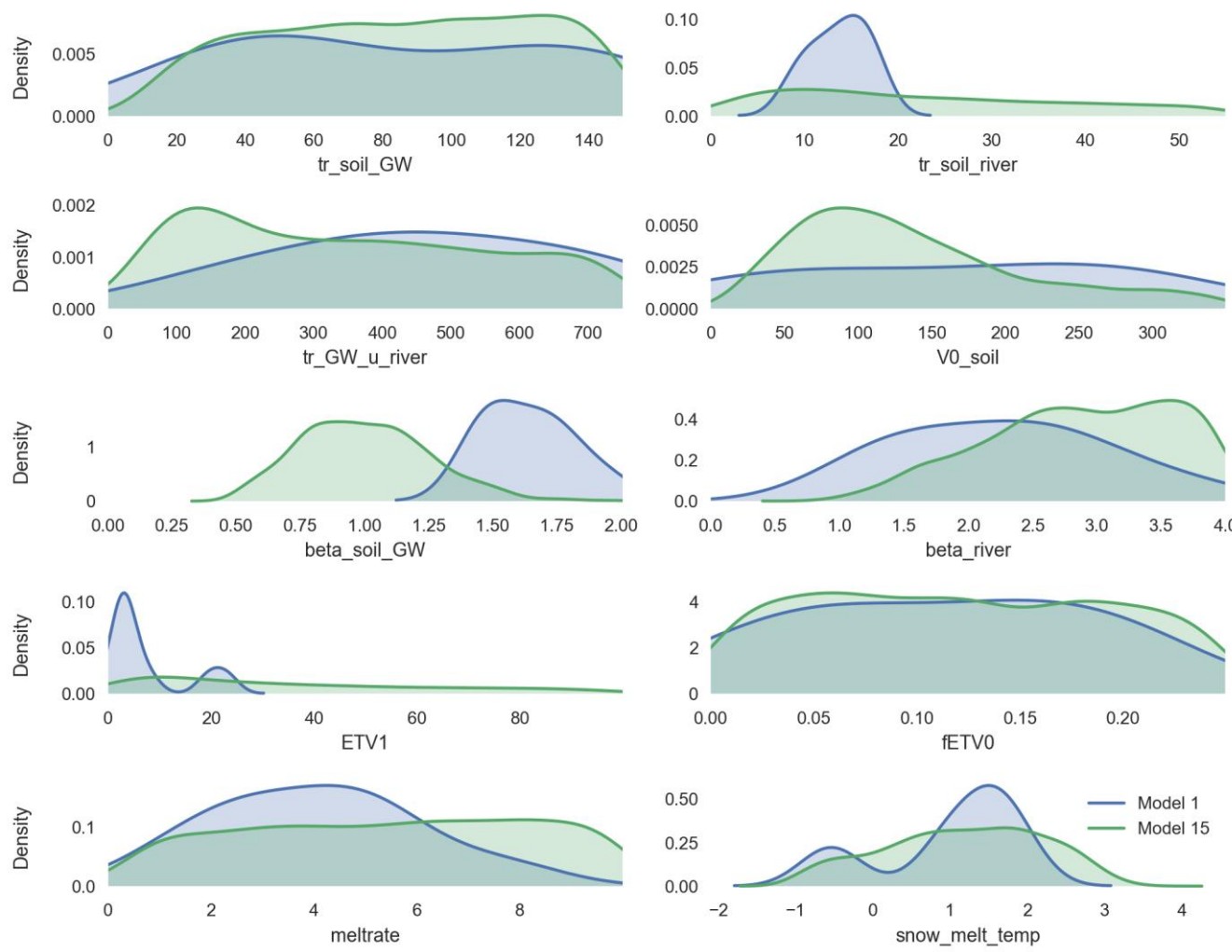

**Figure 8: Distribution of all parameters shared by Model 1 (blue) and Model 15 (green), fitted with kernel density.**

### 3.3 Behavioural runs of HBV-Light

HBV-Light performs best of all models in this study. Its performance increases from the calibration to the validation period, especially in regard of the maximal values of the objective functions (Figure 5, Figure 6). The largest differences manifest in the values for the RSR and the NSE between HBV-Light and the other models. Contrastingly, HBV-Light seems to have problems in simulating the base flow of the Fulda catchment in the calibration period, resulting in a worse value for the logNSE in comparison to Model 15. However, in the validation period HBV-Light performs better in regard to the logNSE. In addition,

HBV-Light has a very wide range for the values of the objective functions in the validation period, hinting to a large parameter equifinality.

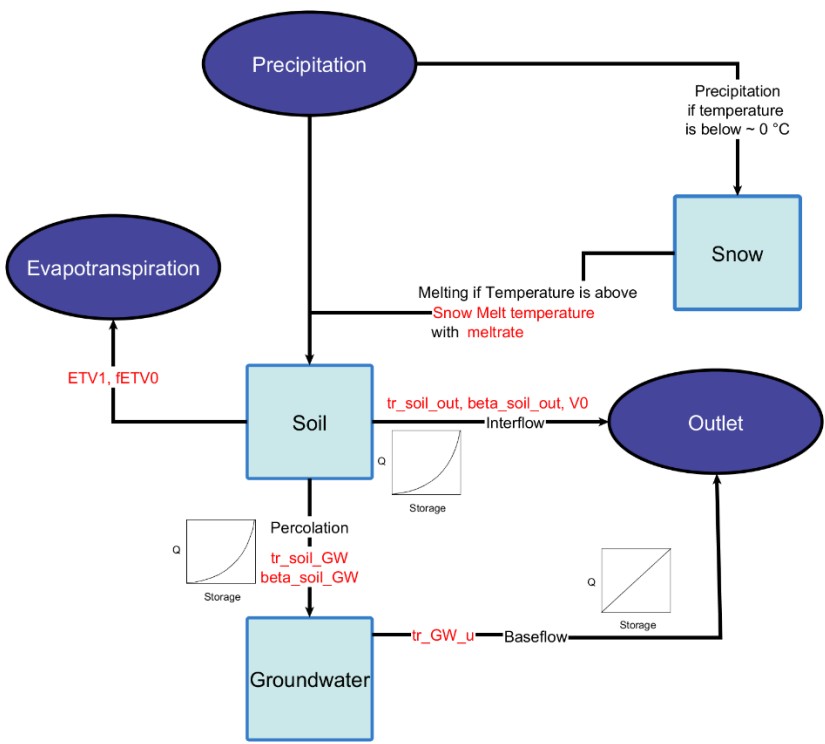

**Figure 9: Structure of the final Model 15 with water storages (light blue), boundaries (dark blue), temporary storages (white) calibrated parameters (red), fluxes (black arrows) and flow curves (for all applicable fluxes. Water reaching the outlet is shifted one day into the future.**

## 4 Discussion

### 4.1 Overview

The results show that Model 1 fell short on simulating the catchment correctly. Mainly caused by a slow reaction to precipitation events, which reduced discharge peak prediction and caused the model to focus on evapotranspiration to handle the excess water. Still, it was a good basis to determine relevant processes by incremental model breakdown. Insights from this led to an improvement in the performance of Model 15, while at the same time allowed a reduction of the number of

parameters (from n=19 to n=10)(Table 1). Nevertheless, even this improvement did not allow Model 15 to outperform HBV-Light. Still, this suggests that the method of incremental model breakdown is a good way to improve model performance and reduce equifinality through parameter reduction, which improves the identifiability of the model structure (Ambroise, 2004).

It enables insight to which processes are important for discharge simulations in a given catchment. It also allows revealing errors, using them "as a means of discovery" of false model assumptions (Elliott, 2004). It should be noted that this method does not necessarily lead to an improved model performance, but it allows creating a model, which relies on fewer processes, parameters and assumptions, thus being an application of "Occam's razor" (Clark et al., 2011).

## 4.2 Inspection of internal processes

Even though Model 1 did give sufficient but not excellent results, it was a good foundation for the construction of Model 15. Due to the implementation of many processes in Model 1, all those processes could be examined on their effect on the simulation. Upper groundwater, evapotranspiration and the simulation of the snow storage and snowmelt were identified as the most important processes, as for example models 2, 5 or 10 could not achieve behavioural runs without those processes (Figure 5, Figure 6). Model processes of drinking water and the canopy showed only minor impact on model discharge simulation performance (Figure 5, Figure 6). Improved values for the objective function were found for models 4, 6 and 13 with no river, no surfaces and no lower groundwater, as those processes likely hindered the models from being better. Excluding these processes make the models react slower and with this, more accurate to precipitation inputs. The drinking water storage's minor influence might simply be due to the rather low population of 159 persons per km² in the region, and neither water withdrawal for irrigation nor water-consuming industries are relevant players in the region's water cycle. The canopy, however, is commonly regarded as an important factor, as interception can cause 25 % and more of the rainfall not to reach the ground (Link et al., 2004). However, Model 15 was able to get better values for the objective functions than Model 1 even though the canopy was disabled (Figure 5, Figure 6). Seibert and Vis (2012) showed before that canopy storages are not very important for humid climates, as in our case. However, Fenicia et al. (2008) found that this is not the case for dry environments.. This can be seen as an example on how incremental model break down produces models that are adopted to the environment of model testing. Also, the current implementation of the canopy assumes a fixed canopy storage for the whole year. A more realistic approach should be implemented for future applications, as was for example realized in plot scale CMF application coupled to plant growth models for winter wheat (Houska et al., 2014) and perennial grassland (Kellner et al., 2017).

The river storage is most likely too small to be an important reservoir in comparison to the catchment. The surface storages probably do not contribute to the runoff itself, because the catchment is mostly vegetated, which impedes overland flow. Lower groundwater was included because of the ability of the sand- and limestone in the catchment to store large quantities of water and because of the tritium based tracer experiments of Wittmann (2002). He found two distinct groundwater aquifers in the catchment. Their study comes to the results that the lower one of the aquifers must be very large. However, our posterior parameter boundaries indicate a very slow response.

Model 15 falls short in predicting peak flow in summer (Figure 7). Due to that, the model has too much water and needs to compensate for this by overestimating baseflow and evapotranspiration (Figure 7). The problem of not predicting the peak flow in spring completely right is probably caused by the lumped and simple implementation of snowmelt. Most of the snow

in the Fulda catchment is stored in a small area along the ridges, while the lumped model does not make such a spatial distinction. A further, possibly influential, discrepancy between our lumped modelling assumptions and reality is that the snowmelt occurs evenly distributed over the whole catchment, so that the complete snowmelt in the model takes only a few days, often even in only one day. It might also be linked to the evapotranspiration. In times of low evapotranspiration, the water is forced to leave the model as discharge. Therefore, large precipitation events are directly transferred to large peaks. During times of high evapotranspiration, much water can be released into the atmosphere, and as the water in the soil storage of Model 15 flows proportionately more if the storage is already high, this allows the water to stay longer in the soil, which in turn allows more evapotranspiration.

The fluxograms showed that Model 1 did not use the drinking water storage and used the canopy storage only rarely. These observations underline the demand of Clark et al. (2011) that the internal procedures of a hydrological model should be inspected as well, to better understand its functioning. The fluxograms helped to detect that canopy and drinking water are not used by the model.

When examining the median model performance of all reduced models and the resulting Model 15, one can see that the median values of the objective functions of Model 15 are similar to those of Model 13 (Figure 5, Figure 6). Model 15 is considered to be the better representation of the catchment than Model 13, as it has a more streamlined structure and seven parameters less. In addition, the good values for Model 13 are mainly caused by the low number of behavioural runs (n = 2), allowing one very good run to distort the results. Also, Model 15 reaches higher maximal values for the objective functions.

This improved performance of Model 15 in comparison with the Model 1 is overshadowed though, by the higher equifinality of some parameters in Model 15 (Figure 8). In Model 1 for example the parameter EVT1 has two very distinct peaks, while Model 15 distribution for this parameter is spread out widely. The behavior of ETV1 might also be linked to the rightward shift of the parameter beta_soil_GW. This parameter controls the speed in which water leaves the soil in the direction of the groundwater. The increase in its value lets the water stay longer in the soil storage, allowing more Evapotranspiration, which in turn allows the parameter ETV1 be handled more flexible by the model. This might indicate that the parameters of Model 15 need to compensate for the simpler model structure in comparison with Model 1, by having a wider range of possible values. This would allow a single parameter to express the behavior of several processes at once.

## 4.3 Comparison with HBV-Light

All three models show a distinct behavior (Figure 5, Figure 6), with HBV-Light and Model 15 behaving rather similar. The main differences between the models are the ability to predict the peaks, an over/underestimation of base flow and the shape of the hydrograph in general. Model 1 seems to be able to get the shape of the hydrograph during low flow correct (Figure 7), while HBV excels at simulating the peaks. Model 15 is somewhere in between. Those differences are probably caused by the number of storages in the models and processes that mimic saturation excess.

Model 15 and HBV-Light are quite similar with regard to their model structure and the considered hydrological processes. The main differences in model performances is the way the mathematical process descriptions are implemented. HBV-Light has a maximal value for percolation and the triangular weighting function that changes the shape of the flow curve (Seibert and Vis, 2012) .With the maximal value for percolation, additional water is forced to become discharge, as there is no other way it could go. This allows HBV-Light to forecast the peaks better, but also might make the model react too quickly. This behavior though is counteracted by the triangular weighting function of HBV-Light. In contrast, Model 15 predicts the peaks correct only during times of low evapotranspiration. Another main difference exists for the simulation of base flow. Model 1 depicts a highly correlated base flow to the observed one, but the model is overestimating the total amount. Model 15 and HBV-Light mimic the shape and timing of the low flow worse, but predict the amounts better. One reasons for this behaviour might be that a model needs a good representation of the groundwater to simulate discharge minima (Plesca et al., 2012), which  is the case for Model 1, but only to a lesser extent for HBV-Light and Model 15. Those findings highlight the problems occurring when a complex model is reduced in its complexity. Therefore, even if the simpler models are able to get higher values for the objective functions they seem to lose some other traits (like the ability to predict the correct shape of the low flow in this case) in an unpredictable manner.

## 4.4 Does model incremental breakdown allow the construction of improved models?

The improved performance of Model 15 shows that a priori model selection can raise problems, as the models with different process implementations deliver very different results. Those differences cannot be explored if only one predefined model is used. This is in line with the findings of Ley et al. (2016), who used predefined model structures on a large amount of different catchments and found that no model was able to simulate all catchments well. Similar results were also found by Kavetski and Fenicia (2011) and Fenicia et al. (2014), who showed that lumped models need to be tailored for single catchments as they are often over-simplified. Therefore, starting from a complex model, simplifying it only where justified, could help resolve the issue of oversimplifications in lumped models.

Lumped models have the advantage of an easy set-up and low data requirements, but this comes at the cost of not being able to address the spatial heterogeneity of the catchments (Ley et al., 2016) and that the parameters and structures have no direct equivalence in the real world (Bergström and Graham, 1998). Therefore, a lumped model structure might simply have been too simple for the upper section of the Fulda, calling for a semi-distributed or even distributed model set up. This is also hinted by a study by Fink and Koch (2010), who where able to model the Fulda Catchment quite well with a modified semi distributed version of SWAT. Overall, we think that the proposed method of incremental model breakdown led to an improvement in model performance. In addition, the model complexity and amount of parameters and with this equifinality were reduced. In this regard, the incremental model breakdown is different to methods like sensitivity analysis where the model structure is untouched, as we reduce the structural model complexity. Both topics are often stated as the main goals of model development e.g. Efstratiadis and Koutsoyiannis (2010) and Gupta and Nearing (2014).

Incremental model breakdown bears, as any model intercomparison study of calibrated models, a risk of overfitting. In the context of this study, overfitting would results in the acceptance of a process that seems only by chance relevant in the calibration period, but has only weak predictive power. Another overfitting effect would be a preference of parameter rich models. An indicator for overfitting are great results in the calibration period but flawed results during validation. This shows

the importance of a validation period that is never used in any selection process, neither for structure nor for parameters. In this study, the performance of the models during validation generally exceeded the performance of the calibration period, despite the different characteristics of those periods. For future studies, longer data time series should be split into three independent periods: (1) parameter calibration period, (2) model selection period and (3) the a posteriori selection process validation period. A second effect when both structural and parameter uncertainty are to be compared, we are not only facing

an equifinality of parameter sets but add equifinality of structures. We based the recognition of relevant processes on the rejection and not the optimization of certain model structures, as suggested by Beven (2006) to gain a robust method. All in all, incremental model breakdown and inspection of parameter distribution, as well as comparison with already established models and the flowpath in a model with a fluxogram might help determine if models do the right things for the right reasons.

## 5 Conclusion

This study shows that the process-based incremental breakdown of a hydrological model using fluxograms and a multi-objective calibration allows the identification of important hydrological processes in a model and the reconstruction of the starting model structure to a more efficient version. We conclude that the method provided offers a useful approach in the identification of relevant hydrological processes. Model frameworks such as CMF facilitate the development of such an approach.

We acknowledge that this study might have led to different results, if a different time step had been used. However, this does not impair the use of the method of incremental model breakdown, as it is meant to find the most important process in a given model set-up and we assume that this mainly depends on the way the model is constructed and not necessarily on the temporal resolution of the input data, at least for the catchment size we investigated in this study. Nevertheless, we recognise that this is a potential direction for future research. Sikorska and Seibert (2018) recently showed that data of sub-daily resolution

become relevant to achieve acceptable model performances with decreasing catchment sizes of 200 km² and smaller. The incremental model breakdown can be used best in two cases: (1) finding out why an existing good model does produce good results in the sense of a diagnostic tool to assess model structures; or, as in this study, (2) determining which processes are most relevant, to allow the streamlining of a model.

One goal of this study was to find another strategic way to test the multiple implementations of catchment functioning. We

were able to distinguish between unnecessary and relevant model processes. Further, it became clearer what causes those problems, by examining the model piece by piece as proposed by Clark et al. (2016). Therefore, this method can be seen as a useful third way, in addition to step-wise model building (Bai et al., 2009; Westerberg and Birkel, 2015) and the comparison

of predefined structures (van Esse et al., 2013; Kavetski and Fenicia, 2011), to explore the realm of multiple hypotheses and helps to keep an open mind, which is needed when going from a complex model to simpler one. (Bergström, 1991). We propose future research should consider an automatic assemblage of model structures to test not only a manually manageable number of models but rather scan a larger variety of feasible combinations, which in turn would allow a completely exhaustive exploration of the space of possible model structure.

*Data availability*. Datasets are available by contacting the Hessian Agency for Nature Conservation, Environment and Geology (HLNUG) (https://www.hlnug.de/service/english.html).

*Competing interests*. The authors declare that they have no conflict of interests.

*Acknowledgements*. We thank the "Hessisches Landesamt für Naturschutz, Umwelt und Geologie" for providing the meteorological and discharge data and Albrecht Weerts, Lieke Melsen, François Anctil and one anonymous referee for their valuable comments which allowed to improve this paper substantially.

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
