# Peer review of "Incremental model breakdown to assess the multi-hypotheses problem"

_Hydrology and Earth System Sciences, 2017_

## Referee Comment (RC1) · L.A. Melsen (Referee) · 27 Dec 2017

Jehn et al. provide a case-study of incremental model-breakdown; starting off with a (benchmark) model including a high number of processes (and parameters), the model is compared to models where fewer processes are explicitly represented. Finally, based on this information, a simplified model is presented with a higher model performance than the benchmark model.

The manuscript is well written and well-structured, and the figures and tables are to the point. I liked the fluxogram. There are, however, some questions, especially about the rationale, that I think need to be addressed, and the results and discussion sections are limited. This can improve with a more in-depth analysis of the results, for which I

provide a (first) suggestion.

About the rationale:

In the introduction and the conclusion the 'incremental model-breakdown' is presented as an alternative next to step-wise model building and comparison of pre-defined structures.

1) My intuition would be to conduct a sensitivity analysis, and based on that determine which processes are relevant and which are not. What is the advantage of doing the incremental model-breakdown rather than a sensitivity analysis? (except that the parameter is completely removed from the model rather than fixed at some point).

2) Another alternative, besides the incremental model-breakdown, step-wise building, and pre-defined structures, is to replace formulations of certain processes with alternative formulations, for example the SUMMA framework which you cite (Clark et al, 2015ab). How does the incremental model breakdown compare to this approach?

3) What is the added value of the incremental model-breakdown compared to all the alternatives? p.2,l.28 states that only a minor quantity of the vast space of possible model structures is explored, but isn't this also true for the incremental model-breakdown as presented in the manuscript, since only a single 'complex' model was employed?

Main points:

The model was run with a daily time step for a catchment in the order of 3000 km2. As becomes clear later on (section 2.5), the response time of the catchment is less than a day. How do you expect this influences your results? Obviously, this temporal resolution is not sufficient to capture the dynamics of the catchment. (follow up on that; It is unclear to me why you had to move the time-series; the river-part could easily be implemented as a routing with a time delay rather than a storage-system, which is more common for rainfall-runoff models).

The discussion of equifinality in the manuscript seems incosistent. Generally, the risk

on equifinality is higher with more degrees of freedom (more parameters compared to the information in the available data for calibration). But on page 13, l.3 is written: '[incremental model-breakdown]..have a positive impact on model performance, given the increased number of behavioural runs'. So; more behavioural runs is positive? But also an implication of equifinality? On p.14, l.11 it states '[incremental model-breakdown].. is a good way to improve model performance and reduce equifinality'. Please clarify.

This relates to my next point: is it a fair comparison to take the mean of the behavioural runs? I have not figured it out myself completely yet, but I don't see why a particular model should be 'punished' for having more (or less) behavioral parameter sets, see e.g. p.16, l.3-7. Perhaps consider another metric to compare the models.

p.9, l.20; for every model, a LHS of 300.000 is taken, despite the number of parameters. So, for models with fewer parameters, each parameter is sampled more often. This could explain why the more frugal models (fewer parameters) have more behavioral runs. Do you think this is the case?

Please add a motivation why you chose these three objective functions. None of the objective functions focusses on high flows, but still peaks and high flows are continuously discussed in the results and discussion section (e.g. p.13,l.17), while low flows are not discussed at all.

Can you provide an order of magnitude for the drinking water abstraction? The process is included in the model because water is abstracted for 80,000 inhabitants (p.4,l.2) but turns out to be unimportant, possible because of low population (159 persons per km2, p.15,l.10). In other words: where did you base the min and max parameter boundaries for drinking water extraction on? (Table 3)

In general, please provide references or motivation how and why you defined these boundaries for your parameters (Table 3).

To continue on that, I would also like to suggest for further analysis; why not showing the distribution of the parameters for the different model formulations? I would be interested to see if any of the parameters is taking over the job of one of the parameters that has been left out. This would result in a shifted parameter distribution. If I may undisclosed refer to my own work; see figure 8 in https://doi.org/10.5194/hess-20-2207-2016 Then, it would be interesting to see which parameters compensate for which processes.

The manuscript lacks a discussion of how the calibration period relates to the validation period. More parameters could fit better in the calibration period but can be flawed in the validation, which is something that should be discussed in relation to model complexity and number of parameters (see Kirchners paper on being right for the right reasons).

Other points:

Please mention the model time step and the temporal resolution of the input data in the 'model input and validation data' section.

Please check the units of Eq.1. V0 is a volume (p.4,l.24) but has the units of a rate. What are the units of V and Q?

Calculating the mean for a NSE is tricky since the NSE is highly non-symmetrical (+1 to minus infinity). Consider using the median.

Figure 3, caption. I think the word 'uncertainty' in 'uncertainty of the behavioural model runs' is not in place here. All you look at is the spread in your behavioural runs, which is certainly different from uncertainty.

The same holds true for p. 17, l. 1, 'less uncertain'

p.2, l.21 comparability -> comparison

p.4, l.6 unnecessary brackets around CMF, 2017
p. 10, table 3; caption; 'indented', in the table: 'intendent' -> intended

p. 16, l.8-13 repetition of p. 15, l.26

---

## Referee Comment (RC2) · F. Anctil (Referee) · 5 Jan 2018

In this paper, submitted by Jehn et al., a breakdown approach is proposed in order to simplify a complex model into a structure with "improved model performance, less uncertainty and higher model efficiency" (line 17, page 1). The method is validated on a 3-year time series from a single gauging station in Germany.

General comment:

The main argument in favour of experimenting with the proposed incremental model breakdown is that it may lead to a better model than the more common stepwise bottom-up approaches, arguing that "there is a chance that they have missed an even better model performance by including further modifications" (line 28, page 2). Yet no

comparison with a stepwise model building is presented, providing no evidence that a breakdown approach is superior.

Major comments:

There is possibly some confusion on the size of the watershed, which drains only about 3 km2 according to line 14, page 3. It is more likely that the size be 2977 km2 and not 2.977 km2, in order to accommodate 108 meteorological stations and an altitudinal range from 150 to 950 m a.s.l. A map of the watershed would have allowed to clarify this issue. It is recommended to add one.

Lumped hydrological models often need shorter time series for calibration than distributed ones. But in the context of a research on the selection of structural components, I am surprised that only 6 years of data was selected for calibration and only 3 more for validation (line 3, page 9). This needs to be justified. Longer series offer the advantage of stabilizing the results in regards to climatological variability. Were there no data available after 1988? At least, the authors need to inform on the climatology of the calibration and validation datasets in regard to the general, say, 30-year climatology. For instance, models usually work much better in wet years than in dry years. Was it the reason for selecting observations from the 80's? The authors should also avoid vague statements like "climatic conditions during the calibration (1980-1985) and validation period (1986-1988) were rather similar" (line 31, page 3). Chances are that they are not so similar at least in terms of low flows, otherwise how can one interpret the raise in validation logNS values in Table 2, in comparison to their calibration counterpart.

The issue of shifting the simulated discharge one day into the future to improve overall performance (line 8, page 9), thus simulating Q(t+1) instead of Q(t), typically falls from some failure in the routing components of the model and sounds more like fudging than modelling. What is the operational consequence of that trick? The argument that rainfalls occur in the "later time of a day" is weak and needs to be substantiated. This

information should be included in Figures 1 and 4.

GLUE is a convenient tool to assess the level of the parameter uncertainty of a model and to identify a number of equifinal (behavioural) parameter sets. Its use here as a calibration tool needs to be better justified (line 13, page 9), for example in comparison to more operational calibration schemes. Here, models variants are essentially compared in Table 2 on the basis of their number of behavioural runs that surpass three thresholds advocated by Moraisi et al. (2007), while parameter uncertainty is not explored. In practice, this has two limitations. 1) No performance information is provided for models 2, 3, 5, 8, and 10, for which the suppression of a structural component turned out detrimental. The issue is that we are provided no information on how much detrimental this operation is, which is quite important to the manuscript since model 15 is essentially built around them. 2) A small gain in performance may lead to a large increase in the number of behavioural runs.

Information in Table 2 is not that informative because it reflects only the behavioural runs. For instance, we are told that model 13 should be dismissed even if its metrics are better than model 15, because of a much lower number of runs to compute metrics (line 4, page 16). It would be easier to address that by giving all the information (not just the mean and the standard deviation) for example in the form of a box plot. From an operational point of view, hydrologists are looking for the best possible model, and variant 13 may fit their needs better than variant 15.

Minor comments:

Are the authors aware of any other hydrological studies on the same site that could offer some basis of comparison?

Figure 2 is not much useful.

Figure 3 would be more intelligible if it would be split in two: a figure for model 1 and another one for model 15.

[Figure]

[Figure]

---

## Referee Comment (RC3) · Anonymous Referee #3 · 12 Jan 2018

The article presents an incremental model breakdown approach to determine an optimal hydrological model structure for rainfall-runoff modeling. The hypothesis of the authors is that one should start from a model structure that includes all possible processes and that this structure should then be incrementally simplified by successively removing the unimportant processes, i.e. those for which the model performance is not degraded or even improved when they are removed from the structure. The approach is demonstrated on a catchment in Germany.

Though the approach is interesting, I have several concerns about the way it is applied and demonstrated:

- I think that the "one-at-a-time sensitivity analysis" approach that is applied makes the hypothesis that all processes are independent from each other in the model structure.

[Figure]

However, this is probably not the case and it is most likely that there are interactions and compensations between model components. Therefore, I find it is difficult to conclude on the individual value of each component based on these tests only. There is no guarantee that the model structure selected at the end is optimal, since only a very limited number of structures among all the possible ones have been tested. It is likely that there are many options which are close to each other in terms of performance.

- The parameter sampling approach, drawing 300,000 parameter sets for each structure, makes that the parameter space will be much more densely scrutinized in the case of a model with 10 parameters than in the case of a model with 19 parameters. This means that the chance of getting behavioral parameter sets is much more limited in the second case than in the first case. This may induce a bias in the way the models are compared when using the GLUE approach. This should at least be discussed or ideally further tested.

- The way the structures versions are selected is unclear. Is this based on results in calibration or in validation? Actually these two options should be tested and discussed. Furthermore, how a model structure is judged to be significantly better than another? Is there any threshold in model improvement or statistical test associated?

- The robustness of the structure selection should be discussed. The model structure is selected based on the use of the first period as calibration and the second as validation. I think the authors should at least test the procedure by inverting the role of the two periods. It is likely that the structure selection may end up (maybe not on this catchment but there are probably cases where it may happen) with different model structures in the two cases. This raises the problem of equifinality in the choice of model structures, and may be a limit of the proposed approach. The selected structure may be over-specialized for the selection period and not really transposable on periods with other conditions. This is what can be observed in the case of model parameters and it is probably also the case in terms of structures. This is probably even a larger problem for periods with much contrasted characteristics.

- The authors did not really discuss the respective roles of structural and parametric complexity in the results. At the end, they have a much more simple structure than at the beginning... but which still has ten parameters, which may appear as overparameterized at the daily time step. It may be interesting to have even more simple model structures, to see how the further simplification possibly leads to degradation in the modeling.

- The authors criticize the usual approach which takes existing models, with interesting arguments. To further demonstrate the value of their approach compared to the classical one, they could test an existing model (e.g. HBV or another model of this type) as a benchmark, to explain the added value of their approach compared to the case when one simply take an existing model.

- Last, I find that making the test on at least a second catchment with contrasted characteristics may strengthen the conclusions. Here the results may be obtained only by chance. There is no guarantee that the results are general outside this case study.

I also have other comments detailed below. In summary, I think there is valuable material in the article, but that the methodology should be further tested and more thoroughly evaluated to provide a more convincing demonstration of its usefulness. I suggest major revision.

Detailed comments

1. P2,L28: This is probably true for all modeling approaches!

2. Section 2.1: Say in which country the basin is located. Maybe a location map could be added. Is catchment size actually 2.977 or 2,977 km$^2$?

3. P4,L10: I find that the definition of a process in the structure should be given. When a process is removed, what happens in the connections in the structure, especially when there are several branches coming to/departing from this process?

4. P6,L18-19: As mentioned in the major comments above, I think that it should be

explained how a version is considered to be significantly better or worse than another.

5. P9,L3-8: Why there is not a pure time-delay parameter (possibly non integer) in the model that would be added in the model structure to account for this time shift and to make it more generally applicable?

6. P9,L12-14: Please remind in brackets for each criterion the optimal value and range of variations, to avoid misunderstanding in the interpretation of results for readers not fully familiar with these criteria.

7. Table 3: Please add a column for units. Maybe also add a column to remind in which structural element (as defined in Table 1) each parameter is included. In the caption: "all model parameters"

8. P11,L11-12: Is not that expected by construction that all model structures have less parameters than the original one?

9. P16,L8-15: This seems to repeat the last paragraph of the previous page.

---

## Author Comment (AC1) · 21 Feb 2018

We would like to thank the reviewers for their highly constructive comments on the manuscript "Incremental model breakdown to assess the multi-hypotheses problem"

(comments of the referees are printed in blue, responses of authors are held in black, added text to the manuscript is in italic)

**Response letter to Reviewer #1 (L.A. Melsen)**

Jehn et al. provide a case-study of incremental model-breakdown; starting off with a (benchmark) model including a high number of processes (and parameters), the model is compared to models where fewer processes are explicitly represented. Finally, based on this information, a simplified model is presented with a higher model performance than the benchmark model. The manuscript is well written and well-structured, and the figures and tables are to the point. I liked the fluxogram.

We created a citable repository for the fluxogram and its code which is now referenced in the paper (http://doi.org/10.5281/zenodo.1137703).

There are, however, some questions, especially about the rationale, that I think need to be addressed, and the results and discussion sections are limited. This can improve with a more in-depth analysis of the results, for which I provide a (first) suggestion. About the rationale: In the introduction and the conclusion the 'incremental model-breakdown' is presented as an alternative next to step-wise model building and comparison of pre-defined structures.

1) My intuition would be to conduct a sensitivity analysis, and based on that determine which processes are relevant and which are not. What is the advantage of doing the incremental model-breakdown rather than a sensitivity analysis? (except that the parameter is completely removed from the model rather than fixed at some point).

"Incremental model-breakdown" has the same aims as a sensitivity analysis. The main difference is, as you state, that a process (together with its parameters) is removed completely and does not remain in the model anymore within the incremental model-breakdown. We see this as advantageous, as it reduces the structural complexity of the model. To make this clearer we added the following sentence to the discussion: *"In this regard, the incremental model breakdown is different to methods like sensitivity analysis where the model structure is untouched, as we reduce the structural model complexity."*

2) Another alternative, besides the incremental model-breakdown, step-wise building, and pre-defined structures, is to replace formulations of certain processes with alternative formulations, for example the SUMMA framework which you cite (Clark et al, 2015ab). How does the incremental model breakdown compare to this approach?

Step-wise model building has the same goal as the incremental model breakdown: Finding the right model. However, the focus differs. The SUMMA approach (which is entirely possible using CMF as a base framework) deals with the question: "What is the best formulation of a process in a given model?", while the question for the incremental model breakdown is: "What is the best overall structure for a model in a given catchment?". In future studies it would be worthwhile to combine both approaches, to get an even more thoroughly exploration of the catchment. To clarify this, we added the following sentence to the introduction: *Clark et al (2015ab) propose with the SUMMA concept another approach to test multiple hypotheses. Their question is: do we use the right formulation for this process? This study asks instead: Is the process relevant for this catchment at all?*

3) What is the added value of the incremental model-breakdown compared to all the alternatives? p.2,l.28 states that only a minor quantity of the vast space of possible model structures is explored, but isn't this also true for the incremental model-breakdown as presented in the manuscript, since only a single 'complex' model was employed?

We agree that this was ambiguously worded. It is clear that our approach will not be able to sample the space of possible model structures exhaustively. Nevertheless, we think that incremental model-breakdown samples a larger part of the potentially available model

structure space than most other approaches, as we start with a very complex model structure (complex in the realm of lumped models), containing all processes which seem to be important for the catchment and trim it down sequentially. This way, more processes might be considered, as when starting with a simple model structure, adding pieces and settle for a structure once a sufficient value of the objective function is reached. To clarify, we added the following line to the conclusions: *From the surface, the water is either directly routed to the river or enters three serial soil/groundwater layers, which in turn would allow a completely exhaustive exploration of the space of possible model structure.* And the following sentence to the introduction: *While still not being able to sample the entire space of possible model structures, this approach might find some model structures which are likely missed with other methods.*

Main points:
The model was run with a daily time step for a catchment in the order of 3000 km2. As becomes clear later on (section 2.5), the response time of the catchment is less than a day. How do you expect this influences your results?
Obviously, this temporal resolution is not sufficient to capture the dynamics of the catchment. (follow up on that; It is unclear to me why you had to move the time-series; the river-part could easily be implemented as a routing with a time delay rather than a storage-system, which is more common for rainfall-runoff models).
We agree that a daily time step is insufficient to model all subdaily dynamics in mesoscale catchments. However, most of the public hydrological data are only available on a daily time step. Therefore, modellers have to cope with it. One approach is the routing with a time delay, which we considered in model 1 where we included a "river" storage which simulated a behaviour with a retention time. But this showed to be less appropriate, as Model 1 was not being able to produce behavioural runs with this process included. Our approach of shifting the time series by one day is another viable option, see Bosch et al. (2004) or Asadzadeh et al. (2016).
We would like to stress that routing or shifting does not affect the idea of our paper, which is presenting an alternative blueprint for hydrological model set up rather than a case study and best model practice for the Fulda river.

Asadzadeh, M., Leon, L., Yang, W. and Bosch, D.: One-day offset in daily hydrologic modeling: An exploration of the issue in automatic model calibration, J. Hydrol., 534, 164–177, doi:10.1016/j.jhydrol.2015.12.056, 2016.
Bosch, D. D., Sheridan, J. M., Batten, H. L. and Arnold, J. G.: Evaluation of the SWAT model on a coastal plain agricultural watershed, Trans. ASAE, 47(5), 1493–1506, doi:10.13031/2013.17629, 2004.

The discussion of equifinality in the manuscript seems incosistent. Generally, the risk on equifinality is higher with more degrees of freedom (more parameters compared to the information in the available data for calibration). But on page 13, l.3 is written: '[incremental model-breakdown]..have a positive impact on model performance, given the increased number of behavioural runs'. So; more behavioural runs is positive? But also an implication of equifinality? On p.14, l.11 it states '[incremental model-breakdown]. is a good way to improve model performance and reduce equifinality'. Please clarify. This relates to my next point: is it a fair comparison to take the mean of the behavioural runs? I have not figured it out myself completely yet, but I don't see why a particular model should be 'punished' for having more (or less) behavioral parameter sets, see e.g. p.16, l.3-7. Perhaps consider another metric to compare the models.
Our use of the term "behavioural" was indeed inconsistent. We deleted this section and rephrased all other sections where the term "behavioural" was used in this way. We further do not use the number of behavioural runs as a performance indicator anymore. To further increase the quality of the evaluation we now included the NSE as fourth objective function particularly focusing on model performance for higher flows.

 for every model, a LHS of 300.000 is taken, despite the number of parameters. So, for models with fewer parameters, each parameter is sampled more often. This could explain why the more frugal models (fewer parameters) have more behavioural runs. Do you think this is the case?

It is true that the parameter space of the models with less parameters is sampled more exhaustively. Nevertheless, LHS is a robust enough method to counter this. As the parameter space is sampled very uniformly when using LHS, a smaller number of runs is needed, as in comparison with e.g. Monte Carlo Algorithms.

The LHS allows to calculate how many runs are needed for good sampling of the parameter space (see McKay et al. (1979)) and this threshold (n=262,144 for 19 parameters) is achieved for all models. This is also in line with our personal experience when using LHS. Usually, models reach good values for the objective functions in the first few hundred runs (even when they are complex), and all following runs are adding only small increments in performance.

Still, our procedure might allow models with less parameters to get more behavioural runs. Therefore we now excluded the number of behavioural runs as a performance indicator.

> McKay, M. D., Beckman, R. J. and Conover, W. J.: A Comparison of Three Methods for Selecting Values of Input Variables in the Analysis of Output from a Computer Code, Technometrics, 21(2), 239, doi:10.2307/1268522, 1979.

Please add a motivation why you chose these three objective functions. None of the objective functions focusses on high flows, but still peaks and high flows are continuously discussed in the results and discussion section (e.g. p.13,l.17), while low flows are not discussed at all.

We agree and see this shortcoming. Therefore we now included the NSE in our multi objective calibration approach as a fourth objective function. Accordingly, we added respective parts in the methods, results and discussion sections.

Can you provide an order of magnitude for the drinking water abstraction? The process is included in the model because water is abstracted for 80,000 inhabitants (p.4,l.2) but turns out to be unimportant, possible because of low population (159 persons per km2, p.15,l.10). In other words: where did you base the min and max parameter boundaries for drinking water extraction on? (Table 3)

As we did not find reliable data to quantify the influence of the drinking water abstraction, we decided to include a subjective estimation, to test whether there is a potential influence on the water flux estimation and if so, how large this influence is. In the revised version of the manuscript, we state this up-front: "*As the influence of the drinking water abstraction is not known, the amount of water abstracted is calibrated*". As it turns out, drinking water abstraction is of marginal influence in the catchment for the annual water balance.

In general, please provide references or motivation how and why you defined these boundaries for your parameters (Table 3).

We included the following section in the calibration and validation section to explain the parameters and also present units for all parameters in Table 3:

*The lower and upper bounds for V0_soil and ETV1 were taken from Blume et al. (2016) for typical field capacities reported for German soils in the range of 20 to 300. Canopy parameters are in line with values provided by Breuer et al. (2003). Groundwater transit times are roughly corresponding with Wittmann (2002) and Wendland et al. (2011). For all other parameters we could not find reliable data and thus estimated them subjectively. The parameters use a wide range intentionally to allow the parameters to adapt to the very different model structures.*

> Blume, H.-P., Brümmer, G. W., Horn, R., Kandeler, E., Kögel-Knabner, I., Kretzschmar, R., Stahr, K., Wilke, B.-M., Scheffer, F. and Schachtschabel, P.: Kapitel 9: Böden als Pflanzenstandorte, in Scheffer/Schachtschabel Lehrbuch der Bodenkunde, Springer Spektrum, Berlin Heidelberg., 2016.

Breuer, L., Eckhardt, K. and Frede, H.-G.: Plant parameter values for models in temperate climates, Ecol. Model., 169(2–3), 237–293, doi:10.1016/S0304-3800(03)00274-6, 2003.

Wittmann, S.: Tritiumgestützte Wasserbilanzierung im Einzugsgebiet von Fulda und Werra, <http://www.hydrology.uni-freiburg.de/abschluss/Wittmann_S_2002_DA.pdf>, Diploma-Thesis at the Institut for Hydrology, Albert-Ludwigs-University Freiburg, 2002.

Wendland, F., Berthold, G., Fritsche, J.-G., Herrmann, F., Kunkel, R., Voigt, H.-J. and Vereecken, H.: Konzeptionelles hydrogeologisches Modell zur Analyse und Bewertung von Verweilzeiten in Hessen, Grundwasser, 16(3), 163–176, doi:10.1007/s00767-011-0169-6, 2011.

To continue on that, I would also like to suggest for further analysis; why not showing the distribution of the parameters for the different model formulations? I would be interested to see if any of the parameters is taking over the job of one of the parameters that has been left out. This would result in a shifted parameter distribution. If I may undisclosed refer to my own work; see figure 8 in https://doi.org/10.5194/hess-20-2207-2016

This is a good addition to the paper. Therefore, we now created a parameter distribution plot for the parameters shared by Model 1 and Model 15, to enable a more thoroughly comparison. To explain this plot, we added the following sentences to the results:

*The remaining ten parameters in Model 15 behave different from the same ones in Model 1 (Figure 5). Some parameters like tr_soil_GW and fEVT0 have almost the same density distribution. Still, there are several parameters like tr_soil_river and ETV1 whose density is much more focused around a specific value for Model 1 than for Model 15.*

Regarding the discussion:

*Model 1 has less equifinality in some parameters, compared to Model 15 (Figure 5). E.g. the parameter ETV1 has two very distinct peaks for Model 1, while for Model 15 the distribution for this parameter is widely spread. The behavior of ETV1 might also be linked to the rightward shift of the parameter beta_soil_GW. This parameter controls the speed in which water leaves the soil in the direction of the groundwater. The increase in its value lets the water stay longer in the soil storage, allowing more evapotranspiration, which in turn allows the parameter ETV1 be handled more flexible by the model.*

[Figure]

Figure 1: Distribution of all parameters shared by Model 1 (blue) and Model 15 (green), fitted with kernel density.

Then, it would be interesting to see which parameters compensate for which processes. The manuscript lacks a discussion of how the calibration period relates to the validation period. More parameters could fit better in the calibration period but can be flawed in the validation, which is something that should be discussed in relation to model complexity and number of parameters (see Kirchners paper on being right for the right reasons).

How do models compensate lack of realism is indeed a highly interesting question. We think this is better to show exemplary for selected processes in a smaller catchment were the relevant processes are known and merit a study on its own.

The main criteria to determine which processes were important, was the ability of model to have any behavioural model runs at all. To understand the influence of the model structure on the parameters we included Figure 1. We did this only for Model 1 and Model 15, as those are the most important models in the study. To make it more clear on what the process selection was based we added the following sentence to the Material and Methods section:

*The main criteria to determine the value of a process was the ability of the model to produce behavioral runs in the calibration period at all.*

To better explain the differences between the calibration and the validation period we added a cumulative sum plot for the precipitation and discharge (Figure 2). With this it is more clear that both periods are different. In addition to the figure, we added the following text to the Material and Methods section:

*The model time step and temporal resolution of the data are both daily. Both the validation and the calibration period behave differently in regard of their patterns of precipitation and discharge (Figure 1). The calibration period is wetter and contains six of the seven large rainfall events (>30 mm $d^{-1}$). In addition, in both periods there is one year representing contrasting extreme weather conditions. In 1985, during the calibration period, very little discharge is observed with at the same time high precipitation, while in 1988 during the validation period high discharge was recorded at comparingly low precipitation.*

Also we added the following sentences to the end of the discussion:

*Incremental model breakdown bears, as any model intercomparison study of calibrated models, a risk of overfitting. In the context of this study, overfitting would results in the acceptance of a process that seems only by chance relevant in the calibration period, but has only weak predictive power. Another overfitting effect would be a preference of parameter rich models. An indicator for overfitting are great results in the calibration period but flawed results in validation. This shows the importance of a validation period that is never used in any selection process, neither for structure nor for parameters. In this study, the performance of the models during validation generally exceeded the performance of the calibration period, despite the different characteristics of those periods.*

*All in all, Incremental model breakdown and inspection of parameter distribution, as well as comparison with already established models and the flowpath in a model with a fluxogram might help determine if models do the right things for the right reasons.*

[Figure]

*Figure 2: Cumulative discharge plotted against cumulative precipitation for the calibration and validation period and two years with extreme behavior. For the calibration and validation period, the cumulative discharge and precipitation are the average of the corresponding years.*

Other points:
Please mention the model time step and the temporal resolution of the input data in the 'model input and validation data' section.
Added as proposed.

Please check the units of Eq.1. V0 is a volume (p.4,l.24) but has the units of a rate. What are the units of V and Q?
We changed the section to make it more clear. The description of the kinematic wave in CMF is now more exactly described by discarding *tr* and introducing *Q0*, which is the flux in m³ per day when the volume of the storage equals the parameter $V_0$.

Calculating the mean for a NSE is tricky since the NSE is highly non-symmetrical (+1 to minus infinity). Consider using the median.
Changed as proposed. Boxplots use the median now.

Figure 3, caption. I think the word 'uncertainty' in 'uncertainty of the behavioural model runs' is not in place here. All you look at is the spread in your behavioural runs, which is certainly different from uncertainty.
Changed "uncertainty" to" range".

The same holds true for p. 17, l. 1, 'less uncertain'
We realized that this wording is confusing and deleted it.

p.2, l.21 comparability -> comparison
Changed as proposed.

p.4, l.6 unnecessary brackets around CMF, 2017
Changed as proposed.

p. 10, table 3; caption; 'indented', in the table: 'intendent' -> intended
Changed as proposed.

p. 16, l.8-13 repetition of p. 15, l.26
Deleted all repeated sentences.

---

## Author Comment (AC2) · 21 Feb 2018

We would like to thank the reviewers for their highly constructive comments on the manuscript "Incremental model breakdown to assess the multi-hypotheses problem"

(comments of the referees are printed in blue, responses of authors are held in black, added text to the manuscript is in italic)

**Response letter to Reviewer #2 (F. Anctil)**
In this paper, submitted by Jehn et al., a breakdown approach is proposed in order to simplify a complex model into a structure with "improved model performance, less uncertainty and higher model efficiency" (line 17, page 1). The method is validated on a 3-year time series from a single gauging station in Germany.
General comment:
The main argument in favour of experimenting with the proposed incremental model breakdown is that it may lead to a better model than the more common stepwise bottom-up approaches, arguing that "there is a chance that they have missed an even better model performance by including further modifications" (line 28, page 2). Yet no comparison with a stepwise model building is presented, providing no evidence that a breakdown approach is superior.
We do not see the incremental model breakdown as being superior to the other approaches, but more like another way to explore possible model structures. The main difference is that incremental model breakdown tries to explore the model space another way by turning the stepwise process upside down.
A direct comparison of both approaches by the same set of authors would not work, as experience from one approach will inevitably influence the decision of model building during the other approach. For future work, it might be a worthwhile idea to give two separate research groups the same information about a catchment and let them built a model: One group using incremental model breakdown and one group using stepwise model building. Finally, both resulting model structures are compared in their performance and structure. However, for the current work presented here, which focuses on the general idea of incremental model breakdown, such a comparison would go beyond the scope of the paper.

Major comments:
There is possibly some confusion on the size of the watershed, which drains only about 3 km2 according to line 14, page 3. It is more likely that the size be 2977 km2 and not 2.977 km2, in order to accommodate 108 meteorological stations and an altitudinal range from 150 to 950 m a.s.l. A map of the watershed would have allowed to clarify this issue. It is recommended to add one.
This was a typo. The Fulda Catchment is 2977 km2 in size. We now added a map (Figure 1).

[Figure]

*Figure 1: Relief map of the Fulda catchment for the gauging station Grebenau (black border).*

Lumped hydrological models often need shorter time series for calibration than distributed ones. But in the context of a research on the selection of structural components, I am surprised that only 6 years of data was selected for calibration and only 3 more for validation (line 3, page 9). This needs to be justified. Longer series offer the advantage of stabilizing the results in regards to climatological variability. Were there no data available after 1988? At least, the authors need to inform on the climatology of the calibration and validation datasets in regard to the general, say, 30-year climatology. For instance, models usually work much better in wet years than in dry years. Was it the reason for selecting observations from the 80's?

We thank the reviewer for this comment, but we have a different opinion on this point. Indeed, a longer time series contains more climatic variability. However, a good model should be able to cope with climatic variability, as its inner structure should resemble the real processes in the catchment. This viewpoint is also shared for example by Kirchner (2006) or Klemeš (1986).
Uncertainty about rainfall is one of the major sources of model uncertainty. To reduce this uncertainty, we selected the time period with the greatest number of rainfall stations without missing data relevant for our study area. Any longer or later time series would result in a strongly reduced number of stations. To better describe the data we used, a figure on cumulative discharge and precipitation is now included (see also response to reviewer #1). We also added the following sentence: *Still, the precipitation stays in the long term range for this catchment for all years (Fink and Koch, 2010).*
Finally, we would like to add that the objective of this paper is not to find the "best" model for the Fulda catchment in the sense of a case study, but present a new way of model building using a rejectionist approach.

Kirchner, J. W.: Getting the right answers for the right reasons: Linking measurements, analyses, and models to advance the science of hydrology, Water Resour. Res., 42(3), doi:10.1029/2005WR004362, 2006.
Klemeš, V.: Operational testing of hydrological simulation models, Hydrol. Sci. J., 31(1), 13–24, doi:10.1080/02626668609491024, 1986.

The authors should also avoid vague statements like "climatic conditions during the calibration (1980-1985) and validation period (1986-1988) were rather similar" (line 31, page 3). Chances are that they are not so similar at least in terms of low flows, otherwise how can one interpret the raise in validation logNS values in Table 2, in comparison to their calibration counterpart.
We deleted the statement and added additional data and Figure 2 (in response to reviewer #1) to communicate the climatic conditions more clearly. See also replies to the comment above.

The issue of shifting the simulated discharge one day into the future to improve overall performance (line 8, page 9), thus simulating Q(t+1) instead of Q(t), typically falls from some failure in the routing components of the model and sounds more like fudging than modelling. What is the operational consequence of that trick? The argument that rainfalls occur in the "later time of a day" is weak and needs to be substantiated. This information should be included in Figures 1 and 4.
We still think this is a valid method. We now included further information in respective figure captions.
Most of the public hydrological data are only available on a daily time step. Therefore, modellers have to cope with it. One approach is the routing with a time delay, which we considered in model 1 where we included a "river" storage which simulated a behaviour with a retention time. But this showed to be less appropriate, as Model 1 was not being able to produce behavioural runs with this process included. Our approach of shifting the time series by one day is another viable option, see Bosch et al. (2004) or Asadzadeh et al. (2016).
We would like to stress that routing or shifting does not affect the idea of our paper, which is presenting an alternative blueprint for hydrological model set up rather than a case study and best model practice for the Fulda river.

> Asadzadeh, M., Leon, L., Yang, W. and Bosch, D.: One-day offset in daily hydrologic modeling: An exploration of the issue in automatic model calibration, J. Hydrol., 534, 164–177, doi:10.1016/j.jhydrol.2015.12.056, 2016.
> Bosch, D. D., Sheridan, J. M., Batten, H. L. and Arnold, J. G.: Evaluation of the SWAT model on a coastal plain agricultural watershed, Trans. ASAE, 47(5), 1493–1506, doi:10.13031/2013.17629, 2004.

GLUE is a convenient tool to assess the level of the parameter uncertainty of a model and to identify a number of equifinal (behavioural) parameter sets. Its use here as a calibration tool needs to be better justified (line 13, page 9), for example in comparison to more operational calibration schemes.
We used GLUE as it is widely recognized in the hydrological community and gives a clear statement in regard of the model's capabilities to accomplish predefined criteria. As we were not aiming to find a single best parameterization of our models but rather scrutinize the associated parameter space of our models, we still think that GLUE is an appropriate tool for this question. To make this clear, we added the following sentences to the calibration and validation section:
*It should be noted, that other calibration schemes, objective functions and parameter ranges might have lead to different results. However, we are not striving to find the best performing parameter set. Instead, we uses GLUE for the identification of behavioral model runs to evaluate the various model structures.*

Here, models variants are essentially compared in Table 2 on the basis of their number of behavioural runs that surpass three thresholds advocated by Moraisi et al. (2007), while parameter uncertainty is not explored. In practice, this has two limitations. 1) No performance information is provided for models 2, 3, 5, 8, and 10, for which the suppression of a structural component turned out detrimental. The issue is that we are provided no information on how much detrimental this operation is, which is quite important to the manuscript since model 15 is essentially built around them.

To our understanding, the parameter uncertainty of models 2, 3, 5, 8 and 10 is not worth to consider any further. None of the tested parameter sets has achieved the thresholds of the predefined objective functions. So why bother to evaluate these models? Therefore, we applied the SPOTPY software in such a configuration that unbehavioral model runs are not saved for further analyses. With the now included boxplots for all models with behavioural runs it is also more obvious that all models with behavioural runs were a good deal better than those without, were all model runs are below the lower whisker of the boxplots.

2) A small gain in performance may lead to a large increase in the number of behavioural runs. Information in Table 2 is not that informative because it reflects only the behavioural runs. For instance, we are told that model 13 should be dismissed even if its metrics are better than model 15, because of a much lower number of runs to compute metrics (line 4, page 16). It would be easier to address that by giving all the information (not just the mean and the standard deviation) for example in the form of a box plot. From an operational point of view, hydrologists are looking for the best possible model, and variant 13 may fit their needs better than variant 15.

This is a very valuable comment. We therefore deleted table 2 from the paper and added all behavioural models as boxplots (Figure 2, 3). It is now more obvious that Model 15 delivers runs with much higher values for the objective functions than Model 13. Descriptions referencing to table 2 are changed accordingly.

[Figure]

*Figure 2: Boxplots of the objective functions for all models with behavioural runs in the calibration period. The yellow bar marks the median. Number of behavioural runs noted on boxplot.*

[Figure]

*Figure 3: Boxplots of the objective functions for all models with behavioural runs in the validation period. The yellow bar marks the median. Number of behavioural runs noted on boxplot.*

**Minor comments:**

Are the authors aware of any other hydrological studies on the same site that could offer some basis of comparison?

We found one additional conference paper by Fink and Koch (2010) which we added. The only other study that we are aware of by Wittmann et al. (2002) has already been cited.

Figure 2 is not much useful.

This is true. Therefore, we deleted the figure from the paper.

Figure 3 would be more intelligible if it would be split in two: a figure for model 1 and another one for model 15

We see the problem, the referee is mentioning. Having two time series in one plot is always a bit difficult to look at. Nevertheless, we think that it is necessary here to have both models in the same plot, as it allows a better comparison of the two.

---

## Author Comment (AC3) · 21 Feb 2018

We would like to thank the reviewers for their highly constructive comments on the manuscript "Incremental model breakdown to assess the multi-hypotheses problem"

(comments of the referees are printed in blue, responses of authors are held in black, added text to the manuscript is in italic)

**Response letter to Reviewer #3 (Anonymus)**

The article presents an incremental model breakdown approach to determine an optimal hydrological model structure for rainfall-runoff modeling. The hypothesis of the authors is that one should start from a model structure that includes all possible processes and that this structure should then be incrementally simplified by successively removing the unimportant processes, i.e. those for which the model performance is not degraded or even improved when they are removed from the structure. The approach is demonstrated on a catchment in Germany. Though the approach is interesting, I have several concerns about the way it is applied and demonstrated:
- I think that the "one-at-a-time sensitivity analysis" approach that is applied makes the hypothesis that all processes are independent from each other in the model structure. However, this is probably not the case and it is most likely that there are interactions and compensations between model components. Therefore, I find it is difficult to conclude on the individual value of each component based on these tests only. There is no guarantee that the model structure selected at the end is optimal, since only a very limited number of structures among all the possible ones have been tested. It is likely that there are many options which are close to each other in terms of performance.

We thank the reviewer for the valuable comments on the paper. However, we would like to point out that this paper is meant to introduce a new concept to explore the space of possible model structures. We realize that the method would be validated in a more profound way if
- several more catchments had been used
- the validation and calibration time period had been swapped
- several different time series from the same catchment had been compared
- the incremental model breakdown had been iterated several times, and
- the comparison with other approaches like step-wise model modelling had been more in depth.
We think though that all these suggestions sufficient to fill several papers and would bloat the current paper that is mainly meant for an introduction of a new concept.

We further agree that it is correct to criticize that even in our study only a limited number of models is used. Still, 15 (or 16 with HBV-Light) different models is a large set of different model structures that only few other studies have compared. We therefore think that our work as a good starting point for future, even more comprehensive applications of incremental model breakdown.
We also agree that we cannot ensure independency of all processes from each other and we do not have a guarantee that this process is successful in the end. Failures of this approach would lead either to a model with lower performance than the original in the calibration period or to an overfitted model. While the first type of failure is obvious (we would not submit such a result for publication), the problem of overfitting has been not sufficient discussed in the original manuscript. We added a discussion of overfitting to chapter 4.3. Secondly, the connection of processes is shown in a shift of the parameter space, which we have shown exemplary between model 1 and 15 in Fig. 1. We do not claim , that our method leads to a single optimum model, but we explore a new path to model structure improvement. To clarify this we have extended the last sentence of the introduction with: *…obvious, even if a theoretical optimal model structure is still unknown.*

- The parameter sampling approach, drawing 300,000 parameter sets for each structure, makes that the parameter space will be much more densely scrutinized in the case of a model with 10 parameters than in the case of a model with 19 parameters. This means that the chance of getting behavioral parameter sets is much more limited in the second case

than in the first case. This may induce a bias in the way the models are compared when using the GLUE approach. This should at least be discussed or ideally further tested.
It is true that the parameter space of the models with less parameters is sampled more exhaustively. Nevertheless, LHS is a robust enough method to counter this. As the parameter space is sampled very uniformly when using LHS, a smaller number of runs is needed, as in comparison with e.g. Monte Carlo Algorithms.
The LHS allows to calculate how many runs are needed for good sampling of the parameter space (see McKay et al. (1979)) and this threshold (n=262,144 for 19 parameters) is achieved for all models. This is also in line with our personal experience when using LHS. Usually, models reach good values for the objective functions in the first few hundred runs (even when they are complex), and all following runs are adding only small increments in performance. Still, it might allow models with less parameters to get more behavioural runs, but as we now excluded the behavioural runs as a performance indicator, this does not change the observed performance of the models.

McKay, M. D., Beckman, R. J. and Conover, W. J.: A Comparison of Three Methods for Selecting Values of Input Variables in the Analysis of Output from a Computer Code, Technometrics, 21(2), 239, doi:10.2307/1268522, 1979.

- The way the structures versions are selected is unclear. Is this based on results in calibration or in validation? Actually these two options should be tested and discussed.
To avoid a selection bias, we reserved strictly a period of the dataset that is never used in any selection process. The validation period that is only used as a final check for overfitting. As for any model calibration study, the division of calibration and validation period is in the end arbitrary. We have considered to use multiple calibration periods, but rejected this approach in favour of longer time series. To clarify the meaning of the validation period we included as the second sentence in 2.5: *The validation period is strictly not used in any selection process to avoid overfitting and only used in the last validation step of the overall method*. And further in the chapter we extend: *We used the Generalized Likelihood Uncertainty Estimation (GLUE) methodology (Beven and Binley, 1992) to find behavioral parameters sets for the calibration period.*

Furthermore, how a model structure is judged to be significantly better than another? Is there any threshold in model improvement or statistical test associated?
We realize that is not completely made clear how the model structures were selected. We have not made clear enough that it is not the "good" model we need for the decision, but to collect the reject models. Following other comments related to this, we emphasized this process more in the Introduction:
*A subprocess is marked as necessary, when models lacking it are rejected. On this base, a subsequent model is constructed which uses only meaningful subprocesses. Incremental model breakdown is therefore a rejectionist approach, built on the learning from failure and not an optimization process. Beven (2005 ) assumed that a rejectionist approach is generally better suited to gain insight about process hypotheses.*
and Material and Methods:
*A model is rejected, when it is not able to produce runs of acceptable performance for all parameters. And rejected means in this study, the model is missing a process to important to ignore. If a model lacking a certain subprocess is able to produce behavioural runs that subprocess is irrelevant for this application.*

Beven, K.: A manifesto for the equifinality thesis, J. Hydrol., 320(1–2), 18–36, doi:10.1016/j.jhydrol.2005.07.007, 2006.

The robustness of the structure selection should be discussed. The model structure is selected based on the use of the first period as calibration and the second as validation. I think the authors should at least test the procedure by inverting the role of the two periods. It is likely that the structure selection may end up (maybe not on this catchment but there are probably cases where it may happen) with different model structures in the two cases.

We decided to reserve a strict validation period, never used in any model selection. By switching the periods around the independency of the validation period is lost. However, we discussed the role of the validation period more in chapter 4.3: *Incremental model breakdown bears, as any model intercomparison study of calibrated models, a risk of overfitting. In the context of this study, overfitting would results in the acceptance of a process that seems only by chance relevant in the calibration period, but has only weak predictive power. Another overfitting effect would be a preference of parameter rich models. An indicator for overfitting are great results in the calibration period but flawed results in validation. This shows the importance of a validation period that is never used in any selection process, neither for structure nor for parameters. In this study, the performance of the models during validation generally exceeded the performance of the calibration period, despite the different characteristics of those periods.*

This raises the problem of equifinality in the choice of model structures, and may be a limit of the proposed approach. The selected structure may be overspecialized for the selection period and not really transposable on periods with other conditions. This is what can be observed in the case of model parameters and it is probably also the case in terms of structures. This is probably even a larger problem for periods with much contrasted characteristics.

From the original manuscript it might not be clear, that the validation period is differs slightly from the calibration period. We add, also in reponse to reviewer #1 a new fig. 2 to show the differences between the periods. All models with behavioural runs produce accepted results in the validation period and rejected structures are rejected in the validation period also. We see this as a strong indicator for transferability between time periods, and hope that we clarified this issue with the discussion section above. The GLUE answer to the problem of equifinality is to drop the search for the best model parameterization to accept and instead search for parameterizations to reject. In this study, we transfer this approach to model structures and gain information from the model structures that fail and not from the models that work. The rejectionist approach has been clarified in Material and Methods as given above. Also we are discussing this now in chapter 4.3:

*A second effect when both structural and parameter uncertainty are to be compared, we are not only facing an equifinality of parameter sets but add equifinality of structures. We based the recognition of relevant processes on the rejection and not the optimization of certain model structures, as suggested by Beven (2005) to gain a robust method.*

Beven, K.: A manifesto for the equifinality thesis, J. Hydrol., 320(1–2), 18–36, doi:10.1016/j.jhydrol.2005.07.007, 2006.

- The authors did not really discuss the respective roles of structural and parametric complexity in the results. At the end, they have a much more simple structure than at the beginning but which still has ten parameters, which may appear as overparameterized at the daily time step. It may be interesting to have even more simple model structures, to see how the further simplification possibly leads to degradation in the modeling.

We agree, but see this suggestion as part of future work. This paper is mainly meant to introduce the idea of model breakdown and not to find the "best" model possible. In further studies, it could be tested, to which results it would lead to make several iterations of the incremental model breakdown approach. Testing which processes are the most important for the model, reducing the structure to those processes, define a harder boundary for behavioural runs and repeat the process.

- The authors criticize the usual approach which takes existing models, with interesting arguments. To further demonstrate the value of their approach compared to the classical one, they could test an existing model (e.g. HBV or another model of this type) as a benchmark, to explain the added value of their approach compared to the case when one simply take an existing model.

We now included HBV-Light as a benchmark model and added several sections to explain it.

Material and Methods: *As there have not been many studies regarding the construction of models via modelling frameworks, this study uses HBV-Light as a benchmark to make results more comparable with non-framework studies and to allow a more precise evaluation of the performance of the proposed incremental model breakdown method. HBV-Light is a widely used model, which has proven its functionality in very diverse catchments [Seibert and Vis, 2012]. It is a lumped, parsimonious model. We used the simplest setup of HBV-Light with a single soil storage and no lapse rate. As HBV-Light has no internal way to calculate potential evapotranspiration, we used the same approach by Samani [2000] as for all other models.*

Results: *HBV-Light performs best of all models in this study. Its performance increases from the calibration to the validation period, especially in regard of the maximal values of the objective functions (Table 3, Figure 4). The largest differences manifest in the values for the RSR and the NSE between HBV-Light and the other models. However, HBV-Light seems to have problems in simulating the base flow of the Fulda catchment, resulting in a worse value for the logNSE in comparison to the other models. Here the performance is similar to Model 15. Also, HBV-Light has a very wide range for the values of the objective functions in the validation period, hinting to a large parameter equifinality.*

Discussion: *All three models show a distinct behavior (Figure 5), with HBV-Light and Model 15 behaving rather similar. The main differences between the models are the ability to predict the peaks, an over/underestimation of base flow and the shape of the hydrograph in general. Model 1 captures the shape of the low base flow best, while HBV excels at simulating the peaks. Model 15 is somewhere in between. Those differences are probably caused by the number of storages in the models and processes that mimic saturation excess.*

*Model 15 and HBV-Light are quite similar with regard to their model structure and the considered hydrological processes. The main differences in model performances is the way the mathematical process descriptions are implemented. HBV-Light has a maximal value for percolation and the triangular weighting function that changes the shape of the flow curve (Seibert and Vis, 2012) .With the maximal value for percolation, additional water is forced to become discharge, as there is no other way it could go. This allows HBV-Light to forecast the peaks better, but also might make the model react too quickly. This behavior though is counteracted by the triangular weighting function of HBV-Light. In contrast, Model 15 predicts the peaks correct only during times of low evapotranspiration. Another main difference exists for the simulation of base flow. Model 1 depicts a highly correlated base flow to the observed one, but the model is overestimating the total amount. Model 15 and HBV-Light mimic the shape and timing of the low flow worse, but predict the amounts better. One reasons for this behaviour might be that a model needs a good representation of the groundwater to simulate discharge minima (Plesca et al., 2012). This is the case for Model 1, but only to a lesser extent for HBV-Light and Model 15.*

- Last, I find that making the test on at least a second catchment with contrasted characteristics may strengthen the conclusions. Here the results may be obtained only by chance. There is no guarantee that the results are general outside this case study.

We agree that an additional catchment might yield some interesting results as well. However, as stated in several studies, lumped models need to be tailored for every catchment separately (see e.g. Kavetski and Fenicia (2011) and Fenicia et al. (2014). Therefore, it is to be expected that different catchment would lead to different model structures. Still, we think that this is a worthwhile endeavour for future studies, but would go beyond what can be done for the work presented her.

Fenicia, F., Kavetski, D., Savenije, H. H. G., Clark, M. P., Schoups, G., Pfister, L. and Freer, J.: Catchment properties, function, and conceptual model representation: is there a correspondence?, Hydrol. Process., 28(4), 2451–2467, doi:10.1002/hyp.9726, 2014.

Kavetski, D. and Fenicia, F.: Elements of a flexible approach for conceptual hydrological modeling: 2. Application and experimental insights: Flexible framework

for hydrological modeling, 2, Water Resour. Res., 47(11), doi:10.1029/2011WR010748, 2011.

I also have other comments detailed below. In summary, I think there is valuable material in the article, but that the methodology should be further tested and more thoroughly evaluated to provide a more convincing demonstration of its usefulness. I suggest major revision.

Detailed comments

**1. P2,L28: This is probably true for all modelling approaches!**
This is true and the sentence is removed.

**2. Section 2.1: Say in which country the basin is located. Maybe a location map could be added. Is catchment size actually 2.977 or 2,977 km²?**
We apologize for this error. The Fulda Catchment is almost 3,000 km² large. As proposed, we added a map to make this more clear (see response to reviewer #2).

**3. P4,L10: I find that the definition of a process in the structure should be given. When a process is removed, what happens in the connections in the structure, especially when there are several branches coming to/departing from this process?**
To better explain the removal of processes we added the following sentences to the Material and Methods section: *When a process was removed, the connections leading to it were then connected to the next nearby storage. For example: If the surface storage was removed, the Canopy and the Snow Storage were connected to the soil. Or if the river was removed, all connections leading to it were directly connected to the outlet.*

**4. P6,L18-19: As mentioned in the major comments above, I think that it should be explained how a version is considered to be significantly better or worse than another.**
This comment is dealt with in the answer to major comment.

**5. P9,L3-8: Why there is not a pure time-delay parameter (possibly non integer) in the model that would be added in the model structure to account for this time shift and to make it more generally applicable?**
One approach is the routing with a time delay, which we considered in model 1 where we included a "river" storage which simulated a behaviour with a retention time. But this showed to be less appropriate, as Model 1 was not being able to produce behavioural runs with this process included. Our approach of shifting the time series by one day is another viable option, see Bosch et al. (2004) or Asadzadeh et al. (2016).
We would like to stress that routing or shifting does not affect the idea of our paper, which is presenting an alternative blueprint for hydrological model set up rather than a case study and best model practice for the Fulda river.

> Asadzadeh, M., Leon, L., Yang, W. and Bosch, D.: One-day offset in daily hydrologic modeling: An exploration of the issue in automatic model calibration, J. Hydrol., 534, 164–177, doi:10.1016/j.jhydrol.2015.12.056, 2016.
> Bosch, D. D., Sheridan, J. M., Batten, H. L. and Arnold, J. G.: Evaluation of the SWAT model on a coastal plain agricultural watershed, Trans. ASAE, 47(5), 1493–1506, doi:10.13031/2013.17629, 2004.

**6. P9,L12-14: Please remind in brackets for each criterion the optimal value and range of variations, to avoid misunderstanding in the interpretation of results for readers not fully familiar with these criteria.**
Added as proposed.

**7. Table 3: Please add a column for units. Maybe also add a column to remind in which structural element (as defined in Table 1) each parameter is included. In the caption: "all model parameters"**

We included a column for units as proposed.

This is true. We deleted the sentence.

Deleted all repeated sentences.

---

## Author Response (AR2)

We would like to thank the reviewers for their highly constructive comments on the manuscript "Incremental model breakdown to assess the multi-hypotheses problem"

(comments of the referees are printed in blue, responses of authors are held in black, added text to the manuscript is in italic)

Jehn et al. resubmitted their study on incremental model breakdown. The manuscript has significantly improved compared to the previous version, all the main comments from the reviewers seem to be addressed. I have a few (mainly minor) comments, on interpretation and textual.

Major:

On p8 it is stated that the importance of the process is determined based on behavioural runs in the calibration. As stated further in the manuscript and in my previous review, it would be better to determine this based on the validation period given the risk over overfitting. As is written further in the manuscript, the models mostly improves in the validation, so why not make the validation period decisive in the identification of important processes?

It was noted in the first round of the reviews that our time series of 10 years is rather short. On the other hand, we find a sufficient long validation period, which is never used for any kind of model selection for the final discussion of the results absolutely crucial. Hence we decided to keep a multi year independent validation period but do the model selection process on the explanatory power of the models. For future studies, we will make sure to access longer time series of driving and calibration / validation data. These longer time series will be split in three parts: parameter calibration period, structure selection period and final validation period. To explain this procedure we expanded p8:l21 to:

*For each simplified model, the explanatory model performance during the calibration period was evaluated and the validation period was not used in any model parameter / structure selection process but spared for the  a posteriori model comparison.*

We added to p22:l31:

*[… those periods.] For future studies, longer data time series should be split into three independent periods: (1) parameter calibration period, (2) model selection period and (3) the a posteriori selection process validation period. [A second effect …]*

On p15 and p17, the results refer to the median objective-function values. Although I indeed suggested median rather than medium values, I still doubt whether looking at the median of behavioural runs is providing insights in model behaviour per se. Why not just look at the highest values? Because, as discussed in the previous review, the models with fewer parameters are sampled finer because the total parameter sample remained the same over all models. Theoretically, as also written in your reply, "Still, our procedure might allow models with less parameters to get more behavioural runs", therefore more behavioural runs can also (negatively) influence the median model results while this does not imply that your model got worse. Later on in the manuscript, (p17, l15) you also discuss the max values for the objective function.

We agree that it is difficult to determine which criteria is best for describing the set of behavioural runs. However, the hard criteria for this study is if the models were able to get over the threshold at all. The median is mainly used to compare the models to each other and here we think if captures the

general behaviour of the models best, but as we also discussed the maximal values we think that the behaviour of the models is explained.

To clarify this we added this sentence to 2.5:

*To finally compare the models that were able to produce behavioural runs, we used the median to evaluate the typical behaviour of a run of a given model and the maximal value to determine the best possible performance.*

p.22 l. 10 "The improved performance of Model 15 shows that a priori model selection is not useful, as the models with different process implementations deliver very different results." Although it is true that different models will lead to different results, this sentence is quite a strong statement, as it disqualifies the majority of hydrologic modelling studies in which a single (or even a few) off-the-shelf hydrologic models are used. Although I have the opinion that more attention should be paid to model selection (and therefore I am in favour of modular frameworks), saying that all these other studies are basically 'not useful' would be too strong.

We agree that our wording was a bit harsh. The sentence is changed to:

*The improved performance of Model 15 shows that a priori model selection can raise problems, as the models with different process implementations deliver very different results. Those differences cannot be explored if only one predefined model is used.*

Minor:

p.2 l. 25;

The structure does not need to be static in SUMMA, but is also flexible, as with CMF. I now realize that the methodology in this study can be seen as a methodology / application of dealing with modular modelling frameworks in order to identify the optimal model structure.

To further emphasize the difference between framework and model scrutinizing strategy, we changed the formulation to:

*Clark et al (2015ab) propose another approach to test multiple hypotheses. Here, the number and type of subprocesses stay static, yet the mathematical formulation of the process descriptions are scrutinized by exchange. However, the different model scrutinizing strategies are independent of the selected framework. SUMMA, SuperFLEX, FUSE and CMF should be suitable for incremental model build-up, process exchange as well as our new proposal of the incremental model break down strategy.*

Please check again the units of Eq. 1, it seems that the units of Q0 differ from the units of Q given that V0 is a flux.

$V_0$ was given the unit of a flux by accident, as $V_0$ is a volume. We changed the unit accordingly.

p.8 l. 27 "to[o] important to ignore"

Changed as proposed.

I was expecting to read the calibration and validation period in section 2.2, but they were only mentioned in section 2.5. Perhaps shortly refer to it in section 2.2.

Added as proposed.

Nice Figure 8! It is indeed, as written, against expectation that the parameters are harder to identify for model 15 than for model 1, as model 15 has fewer parameters. Both models have a substantially different number of behavioural runs, which seems to indicate equifinality has not really decreased. What could be possible explanations? Perhaps, some parameters in Model 15 are responsible for several processes at once?

This is a good idea. We added the following sentences to the end of 4.2:

*This might indicate that the parameters of Model 15 need to compensate for the simpler model structure in comparison with Model 1, by having a wider range of possible values. This would allow a single parameter to express the behavior of several processes at once.*

p.20 l. 19, the sentence on canopy seems contradictive (is important, but models without it behave well).

We changed the sentence to make our intended meaning clearer.

*However, Model 15 was able to get better values for the objective functions than Model 1 even though the canopy was disabled (Figure 5, Figure 6). Seibert and Vis (2012) showed before that canopy storages are not very important for humid climates, as in our case. However, Fenicia et al. (2008) found that this is not the case for dry environments. This can be seen as an example on how incremental model break down produces models that are adopted to the environment of model testing.*

p.22. l. 14, the sentence over over-simplified lumped models does not really seem to support your method of simplifying models. Or you could use it as an argument: 'therefore, starting from a complex model, simplifying it only where justified, could resolve this issue' (or something like that).

Added as proposed:

*Therefore, starting from a complex model, simplifying it only where justified, could help resolve the issue of oversimplifications in lumped models.*

Thanks for putting the reviewers in the acknowledgements, but my name is misspelled ;)

We are very sorry about this and hope we did it right this time!

We would like to thank the reviewers for their highly constructive comments on the manuscript "Incremental model breakdown to assess the multi-hypotheses problem"

(comments of the referees are printed in blue, responses of authors are held in black, added text to the manuscript is in italic)

The authors provided a detailed reply to the review comments and made in-depth modifications of the article following the reviewers' suggestions. I think this clarified the main points raised by the reviewers and that the article now deserves publication after considering the minor suggestions detailed below.

Detailed comments: (page and line numbers refer to the pdf version with track changes – "hess-2017-691-author_response-version1.pdf")

1. Abstract: I think the last sentence could more clearly state that the Model 15 did not reach better performance than the benchmark, though it has some merits also (if it is the case, see comments 8 and 9).

To make this clearer we changed the last sentence of the abstract:

*The final model is not able to outperform HBV-Light, but we find that the incremental model breakdown leads to a structure with good model performance, fewer but more relevant processes and less model parameters.*

2. Figure 2: Remind in the caption the starting and ending years of each period.

Added as proposed.

3. P8, L28: "too important"

Changed as proposed.

4. Figs. 5 and 6: Explain in the caption which quantiles of the distribution are shown. To ease visual comparison of performance between calibration and validation, the same Y scale should be used on the two plots of each criterion. Besides, it does not really make sense to show distributions when the sample size (n) is very small (let say below ten). In these cases, I suggest not drawing a distribution but simply showing the individual performances by dots (maybe still highlighting the median dot in yellow).

Changed as proposed.

5. P14, L18: The lower bound of the range for PBIAS should be −100% if I understood well the criterion (?)

Sorry, of course we do not account for negative water. Changed to: "(*optimal value: 0; range: - 100% to +∞)*

6. P14, L19: "constraint"

Changed as proposed.

Changed as proposed.

8. P20, L6-8: Is this comment not erroneous? When looking at the graphs, it seems that HBV-Light has the best median performance on logNSE. Is this comment based on mean performance over all behavioral parameters? In that case, this should be made clear in the text and I suggest adding the mean performance for each model version on top of the graph, above each boxplot.

This comment relates to the performance in calibration where the median of HBV-Light is lower. To make the statement clearer we changed it:

*Contrastingly, HBV-Light seems to have problems in simulating the base flow of the Fulda catchment in the calibration period, resulting in a worse value for the logNSE in comparison to Model 15. However, in the validation period HBV-Light performs better in regard to the logNSE. In addition, HBV-Light has a very wide range for the values of the objective functions in the validation period, hinting to a large parameter equifinality.*

9. P24, L4: Not sure why Model 1 is said to simulate low flows best, since Model 15 and HBV have better logNSE median performance. (see previous comment)

This comment is meant to refer to Figure 7. Here Model 1 can be seen to capture the general shape of the low flows quite good, despite have relatively low logNSE values. To make it more clear we changed it:

*Model 1 seems to be able to get the shape of the hydrograph during low flow correct (Figure 7), while HBV excels at simulating the peaks.*

10. Section 4.3: I find this comparison with the benchmark is interesting since it maybe shows the difficulty of the proposed approach which consists in making a complex model more simple. It reminded me the discussion of Sten Bergström (1991),on model complexity and its role in model development (e.g. p.25 "Going from complex to simpler model structures requires an open mind, because it is frustrating to have to abandon seemingly elegant concepts and theories. It is normally much more stimulating, from an academic point of view, to show significant improvement of the model performance by increasing complexity. Nash and Sutcliffe (1970) presented a strategy for model development from the simple to the complex which may help to avoid this frustration…). Maybe the discussion of the proposed approach could also be put in that perspective.

This is a very valuable source. To address this in our paper we added the following

In the introduction:

[revised manuscript text omitted]

---

## Author Response (AR3)

We would like to thank the editor for his highly constructive comments on the manuscript "Incremental model breakdown to assess the multi-hypotheses problem"

(comments of the referees are printed in blue, responses of authors are held in black, added text to the manuscript is in italic)

Thanks for addressing all concerns by the two reviewers. You are almost there.

I noticed that my comment regarding timestep - Another major concern is the answer to the question related to modelling on daily or sub daily timescales. While in the past, the response provided by the authors may have been valid, this is not the case at the moment (moreover the case study is in Germany with an abundance of data available via DWD and bundesländer).- was not addressed.

I still don't understand why you use dialy timestep (I guess this has to do with the period of investigation 1980s) and/or why the period of the 80s was selected while with current day measurement availability the same study could also be have carried out on hourly or subdaily timesteps (see for instance https://agupubs.onlinelibrary.wiley.com/doi/abs/10.1002/2017WR021201).

The reason we used this time span is that we needed a high density of precipitation stations, as we applied External Drift Kriging (EDK) for interpolation and this method demands a certain density of points to interpolate. In the study area there are not sufficient data and stations at hourly time resolution with enough years of continuous information. In addition, when using EDK the same stations for the whole time period should be used, as an increase of the number of stations of time would introduce a bias in the interpolation. Therefore, the time period was chosen, as it contained the largest number of measuring stations with a continuous time series.

You didnot mention the daily timestep as a limitation of the study in the conclusions. I suggest you add a statement in the conclusions that make this limitation clear and maybe add a statement in the material & methods section why this period was selected

Changed as proposed.

We added the following lines to section 2.2:

[revised manuscript text omitted]

---

## Author Response (AR4)

We would like to thank the editor for his highly constructive comments on the manuscript "Incremental model breakdown to assess the multi-hypotheses problem"

(comments of the referees are printed in blue, responses of authors are held in black, added text to the manuscript is in italic)

Thanks for the quick reply and adjustments.

Please remove "Recently developed method like genRE (van Osnabrugge et al., 2017) would have allowed to run the models with more finely grained data, but were not readily available during the design of this study." and the related reference. It might be better to explain the cause of the decline in precipitation network after 1988 (fall of the Berlin wall in 1989/unification of Germany??).

As my comment was not aimed at boosting my citations and I want to avoid any conflict of interest regarding this point. So if you want to maintain this sentence+reference I want to seek approval from the executive editors.

We are sorry about the inconvenience caused. We did not intend to place you in a conflict of interest. Therefore, we changed the paragraph to:

*The time period from 1979 to 1988 was choosen because External Drift Kriging requires a high density of precipitation stations and this mentioned period covered most measuring stations with continuous data.*

And added the following sentences in the same section:

*The model time step and temporal resolution of the data are both daily, which is sufficient given the size of the catchment and the respective rainfall-runoff response (Sikorska and Seibert, 2018). However, the method presented in this study and the utilized modelling software is not limited to the time step of the forcing or calibration data, but can also be applied on higher resolution data with likely different results.*

Also, we added another sentence to the conclusion to make the intentions of the paper more clear:

*We acknowledge that this study might have led to different results, if a different time step had been used. However, this does not impair the use of the method of incremental model breakdown, as it is meant to find the most important process in a given model set-up and we assume that this mainly depends on the way the model is constructed and not necessarily on the temporal resolution of the input data, at least for the catchment size we investigated in this study. Nevertheless, we recognise that this is a potential direction for future research. Sikorska and Seibert (2018) recently showed that data of sub-daily resolution become relevant to achieve acceptable model performances with decreasing catchment sizes of 200 km² and smaller.*

While discussing these topics in our working group we noticed that the number of measuring stations we stated in section 2.2 as 108 is wrong. 108 is the original amount of stations in our data, but we only used the 51 measuring stations with a continuous time series for the extrapolation. We changed this accordingly. Concerning the "decline" of stations, we would like to state, that we have not checked if the number of precipitation stations went down in whole Germany or if stations have been moved to the former GDR. We just selected the best covered period for the Fulda catchment.

[revised manuscript text omitted]